# Whole genome sequencing and imputation in isolated populations identify genetic associations with medically-relevant complex traits

Lorraine Southam[1,2,*], Arthur Gilly[1,*], Dániel Süveges[1], Aliki-Eleni Farmaki[3], Jeremy Schwartzentruber[1], Ioanna Tachmazidou[1], Angela Matchan[1], Nigel W. Rayner[1,2,4], Emmanouil Tsafantakis[5], Maria Karaleftheri[6], Yali Xue[1], George Dedoussis[3] & Eleftheria Zeggini[1]

Next-generation association studies can be empowered by sequence-based imputation and by studying founder populations. Here we report $\sim 9.5$ million variants from whole-genome sequencing (WGS) of a Cretan-isolated population, and show enrichment of rare and low-frequency variants with predicted functional consequences. We use a WGS-based imputation approach utilizing 10,422 reference haplotypes to perform genome-wide association analyses and observe 17 genome-wide significant, independent signals, including replicating evidence for association at eight novel low-frequency variant signals. Two novel cardiometabolic associations are at lead variants unique to the founder population sequences: chr16:70790626 (high-density lipoprotein levels beta $-1.71$ (SE 0.25), $P = 1.57 \times 10^{-11}$, effect allele frequency (EAF) 0.006); and rs145556679 (triglycerides levels beta $-1.13$ (SE 0.17), $P = 2.53 \times 10^{-11}$, EAF 0.013). Our findings add empirical support to the contribution of low-frequency variants in complex traits, demonstrate the advantage of including population-specific sequences in imputation panels and exemplify the power gains afforded by population isolates.

[1] Wellcome Trust Sanger Institute, Human Genetics, Hinxton CB10 1SA, UK. [2] Wellcome Trust Centre for Human Genetics, University of Oxford, Oxford OX3 7BN, UK. [3] Department of Nutrition and Dietetics, School of Health Science and Education, Harokopio University, Athens 17671, Greece. [4] Oxford Centre for Diabetes, Endocrinology and Metabolism, Radcliffe Department of Medicine, University of Oxford, Churchill Hospital, Oxford OX3 7LE, UK. [5] Anogia Medical Centre, Anogia 740 51, Greece. [6] Echinos Medical Centre, Echinos, Xanthi 67300, Greece. * These authors contributed equally to this work. Correspondence and requests for materials should be addressed to E.Z. (email: Eleftheria@sanger.ac.uk).

Genome-wide association studies (GWAS) for complex medical traits have to date been designed and powered for the discovery of common-frequency variants with small to modest effect sizes. It is becoming increasingly clear that rare and low-frequency variants also play an important role[1]. Utilizing a large reference panel can greatly improve GWAS imputation accuracy[2], capturing a slice of the previously unattainable allelic architecture. Isolated populations can additionally help expedite the identification of low-frequency variants affecting complex traits. The founding event can lead to an increase in allele frequency due to genetic drift, thereby boosting power for GWAS. Here, we use GWAS and exome chip data from the Mylopotamos (MANOLIS) and Pomak villages cohorts as a scaffold (Fig. 1), and impute up to a large reference panel of 5,122 individuals, including 249 MANOLIS samples sequenced at $4\times$ depth. This is the first time WGS data have been generated in this population. We examine 13,541,454 and 15,514,754 single nucleotide variants (SNVs) with minor allele count (MAC)$\geq 2$ in the Pomak and MANOLIS cohorts, respectively, and test for association with traits of cardiometabolic relevance. To enable meta-analysis across potentially related individuals, we implement a method that accounts for non-independence across strata and demonstrate its robustness. We identify eight novel signals for traits of medical relevance.

## Results

**Genetic architecture of Cretan population.** We generated whole-genome sequence data at $4\times$ depth in 249 MANOLIS individuals selected on the basis of genome-wide genotype data to maximize haplotype diversity in the population. To characterize the variation landscape in this isolated population, we aggregated the proportion of SNVs captured across the genome (total $n = 9,554,503$ with MAC$\geq 2$) by functional class (Fig. 2a and Supplementary Table 1) and found that variant densities are inversely correlated with ascribed functional importance. Highest densities are observed in intergenic regions, while coding and splice regions, where disrupting variants may have more severe consequences, are sparsest, in line with observations in other populations[1,3–5]. As expected, we also find that variants with more severe consequences are present in a higher proportion at the lower end of the minor allele frequency (MAF) spectrum compared to the genomic average (Supplementary Fig. 1 and Supplementary Table 2).

Of all autosomal SNVs found in the MANOLIS $4\times$ WGS data 0.52 million (5.81%) were unique compared to the UK10K and 1000 Genomes Project reference panels (Fig. 2b and Supplementary Table 3). Most variants unique to MANOLIS were low-frequency and rare, in fact, the rarer a variant was, the more likely it was to be unique to MANOLIS, with 32% of doubletons being unique. To explore the functionality of these variants, we compared, for each MAF bin, the proportion of unique and shared variants belonging to any given functional consequence in that class, and found that rare variants are more likely to be unique to MANOLIS if they belong to a severe functional class. In particular, we find significant ($P < 1.00 \times 10^{-10}$) enrichment of rare and low-frequency (MAF$\leq 5\%$) coding and regulatory region variants (Fig. 3 and Supplementary Table 4), which is expected when comparing shared, older variants with newer, cohort-specific ones which

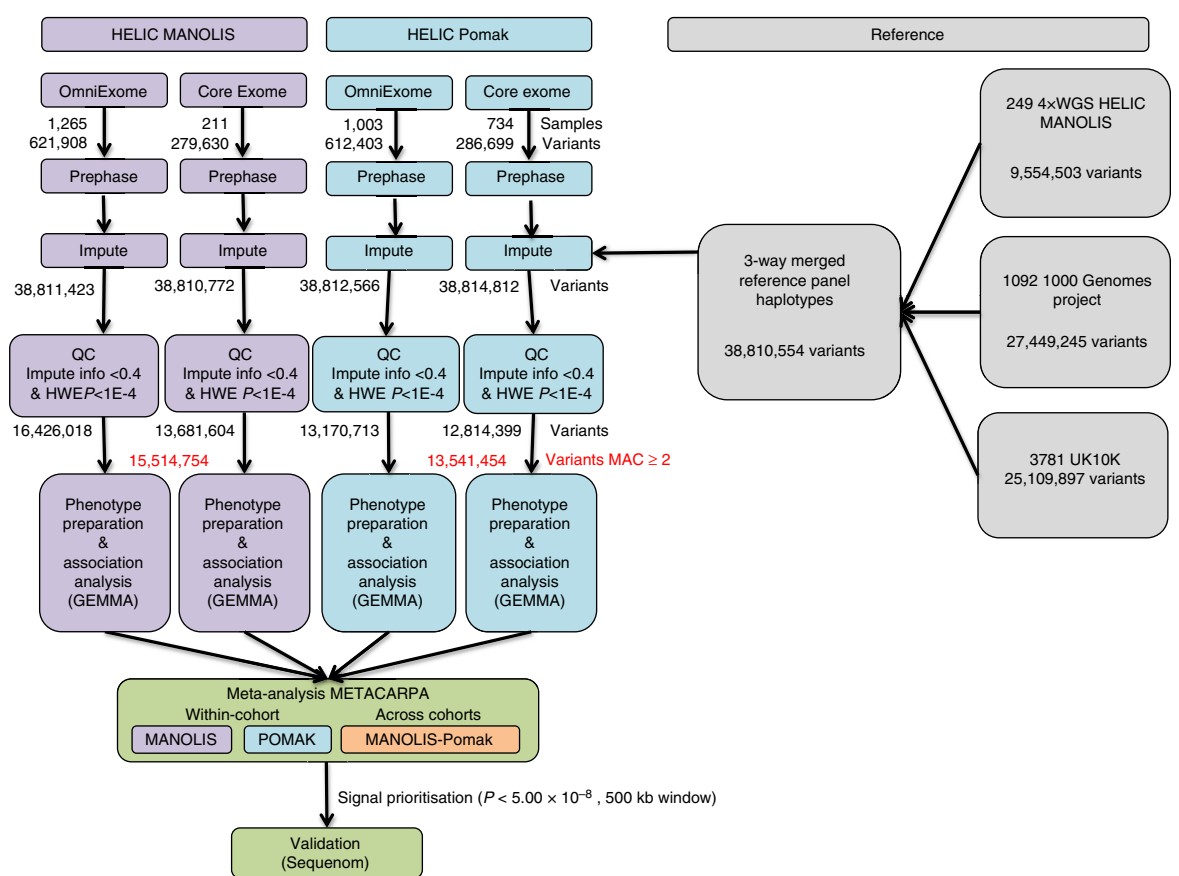

**Figure 1 | Flowchart of study design.** The HELIC cohorts were prephased, imputed and analysed separately by cohort and array, and finally meta-analysed. The variant numbers reported here are total regardless of MAF. Imputed variants are for chromosomes 1–22.

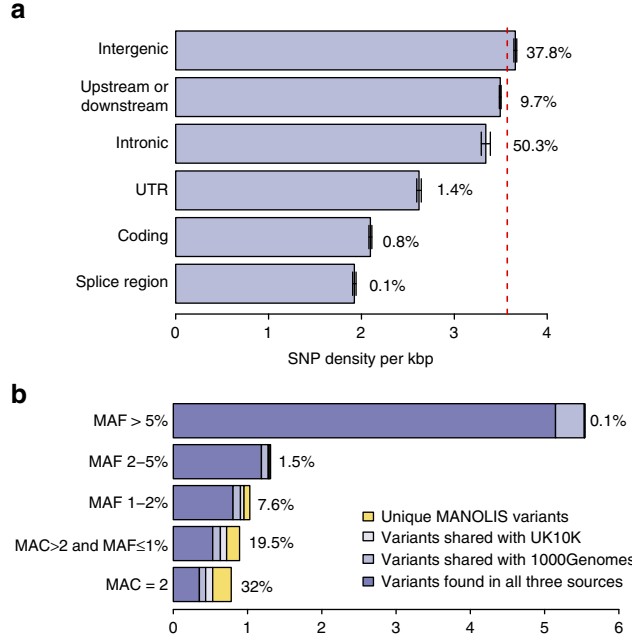

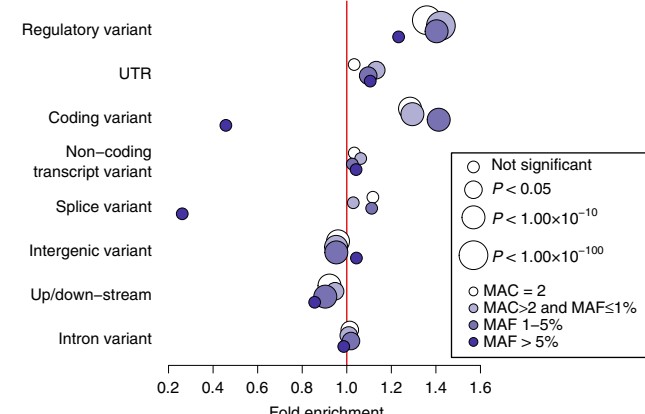

**Figure 3 | Functional enrichment of variants private to the MANOLIS sequences when compared to variants shared with UK10K and/or 1000 Genomes.** Enrichment and depletion of functional classes of variants private to the MANOLIS cohort can be observed in the rare and low-frequency (MAF≤5%), while no significant enrichment is detected in common-frequency variants in any functional class. Numerical values are listed in Supplementary Table 4.

**Figure 2 | Variant sharing and functional annotation. (a)** SNP density per kbp and percentage of total per functional class, based on 9,554,503 variants identified in the HELIC MANOLIS 4× WGS data of 249 samples (MAC≥2). Error bars indicate standard error of the mean; the dashed red line indicates average density genome-wide. **(b)** Variant overlap between 498 HELIC MANOLIS, 7,582 UK10K and 2,184 1000 Genomes Project reference panel haplotypes, by MAF category. Numerical values are given in Supplementary Tables 1 and 2.

haven't yet fully undergone purifying selection. We also find a significant but modest depletion ($P < 1.00 \times 10^{-5}$) of variants annotated as intergenic and upstream/downstream.

**Meta-analysis using METACARPA.** The MANOLIS ($n = 1,476$) and Pomak ($n = 1,737$) cohorts were each genotyped in two tranches (Fig. 1), leading to a requirement for within-cohort meta-analysis. Existing methods to correct for sample relatedness (which is a distinct possibility when meta-analysing within an isolated population) or overlap are based on Pearson's correlation of the $z$-scores[6], but this can lead to overcorrection in the presence of a large polygenic burden[7]. On the other hand, a meta-analysis method that uses tetrachoric correlation[7] and combines $P$ with effect-size based methods[8] can account for non-independence of samples across strata. We implemented the method in openly-available new software, METACARPA (https://github.com/wtsi-team144/metacarpa and http://www.sanger.ac.uk/science/tools/metacarpa). Using simulation, we show that under typical levels of sample overlap (up to 10%) METACARPA reduces false-positive rate inflation by 8%, while conserving power (Fig. 4a and 4b, and Methods). We infer a tetrachoric correlation of 1.96% between $P$ values in the two MANOLIS datasets, and a correlation of 1.84% between the two Pomak datasets. Those values reflect an average within-cohort, cross-dataset kinship of 0.43 and 0.33%, respectively, as measured by pi-hat.

We also compared METACARPA to a genotype-level mega-analysis (Fig. 4a and b, Supplementary Fig. 2). When individual level data are available, a global analysis that takes dataset provenance into account and where overlapping samples are removed maintains the type-I error rate at nominal significance. The power of such a global mega-analysis drops

markedly as sample overlap increases, although it is more powerful than summary-statistic level meta-analyses when no or little overlap is present. When only summary-level statistics are available, METACARPA provides the advantage of a lower false-positive rate than a naïve meta-analysis under typical levels of overlap (0–10%), although it does not control type-I error to nominal levels. Meanwhile, power is conserved compared to the naïve meta-analysis, and is higher than for a sample-level global analysis. As expected, the tetrachoric estimate of overlap is more robust than Pearson's correlation to an excess of signal in the meta-analysed studies (Fig. 4c).

Furthermore, for the HELIC MANOLIS data, we compared the results produced by METACARPA to a mega-analysis of the genotype-level data, as well as a summary-level meta-analysis not accounting for relatedness using the GWAMA software, and found similar median statistics ($\lambda = 0.985 \pm 0.015$) for association with high-density lipoprotein (HDL) (Supplementary Fig. 3). We conclude that all three meta-analysis methods were robust to the moderate levels of relatedness observed between the datasets of the HELIC study.

**Signals associated with traits of medical importance.** We investigated 13 cardiometabolic, 9 anthropometric and 9 haematological traits of medical relevance, and report here genome-wide significant signals ($P \leq 5.00 \times 10^{-8}$) that replicate within (nominal significance and the same direction of effect for each array in a cohort) or across the isolates studied (nominal significance and the same direction of effect in MANOLIS and Pomak). We identify 9 previously-reported GWAS signals (Table 1 and Supplementary Note 1) and 8 novel, internally replicating associations (Table 2, Fig. 5 and Supplementary Fig. 4), which all validate when directly genotyped in the same samples using a different genotyping assay (Supplementary Table 5). None of these novel variants are present in the HapMap haplotypes (http://hapmap.ncbi.nlm.nih.gov), 5 do not have HapMap proxies ($r^2 > 0.8$), and 3 are not present in the 1000 Genomes Project reference panel haplotypes. Three signals were identified in MANOLIS, four in Pomak and one across both isolated populations. Five signals fall just above our Bonferroni-adjusted genome-wide significance threshold ($P < 3.33 \times 10^{-9}$)

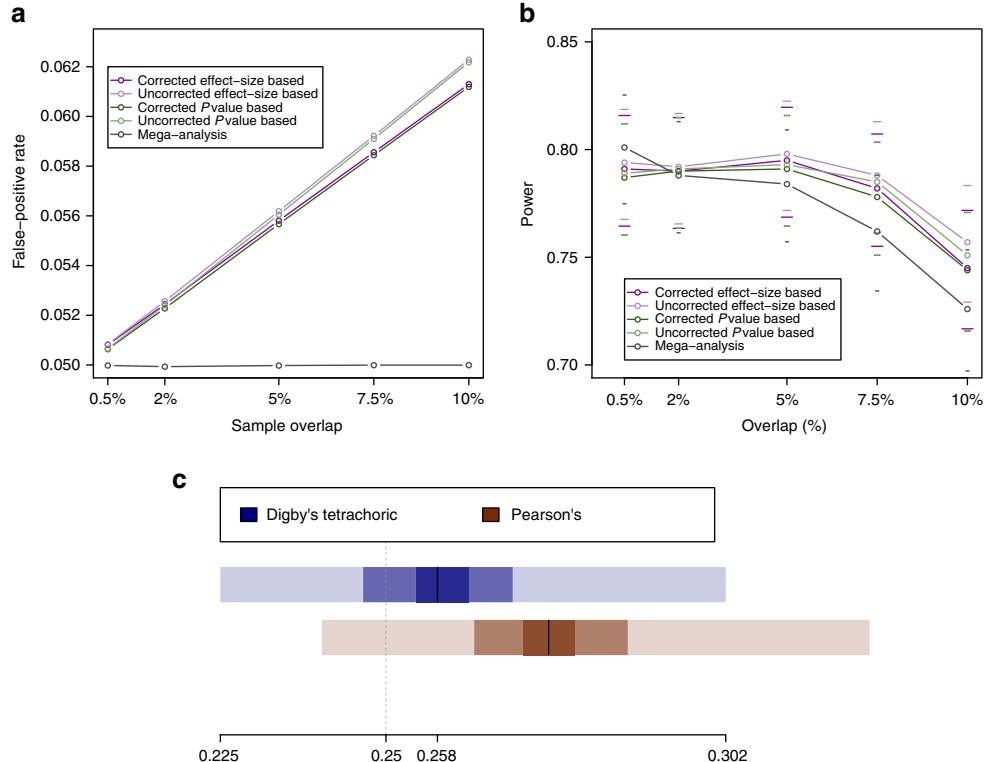

**Figure 4 | False-positive rate and meta-analysis power in the presence of sample overlap using METACARPA.** (**a**) Empirical false-positive rate as a function of sample overlap in 1,000 repeats of a meta-analysis of two studies including 2,000 samples each, at a significance threshold of $5 \times 10^{-8}$. (**b**) Empirical power of the four tests implemented in METACARPA as a function of sample overlap in the same simulation setting. Power is calculated as the discovery rate of a SNP explaining 1% of a standard normal phenotype under the same simulation scenario (for example, a MAF of 1% and an effect size of 0.705, or a MAF of 20% and an effect size of 0.176). (**c**) Compared accuracy of Digby's estimate of tetrachoric correlation and Pearson's correlation for a true (dashed line) 25% overlap under a polygenic burden, with 10,000 SNPs affecting a quantitative trait with 20% heritability. Estimates of correlation for both methods are calculated over 300 genome-wide simulations. The black line indicates the median, shaded rectangles represent the interquintile ranges.

for the effective number of traits tested and are therefore considered tentative. When assessing the fine-mapping potential of these data (Supplementary Methods), we find that the median number of variants in the 95% credible sets is 6.5 and that the median interval length is 546 kbp. This is in line with the expectation of extended LD in founder populations.

We identify a new association with HDL cholesterol at chr16:70790626 (beta $-1.71$ (SE 0.25), $P = 1.57 \times 10^{-11}$, effect allele frequency (EAF) 0.006) (Table 2 and Fig. 5). This variant is present in the MANOLIS sequences only. When MANOLIS sequences are not included in the reference panel a reduced signal is observed at a different variant (Fig. 5c,d). This is the strongest new signal for HDL in MANOLIS and explains 3.24% of the phenotypic variance. Chr16:70790626 resides in intron 11 of the *VAC14* gene. The encoded protein is involved in the regulation of phosphatidylinositol 3,5-bisphosphate levels and the biogenesis of endosome carrier vesicles[9,10]. In animal models, knocking out *Vac14* causes death within 2 days of birth caused by the disruption of phosphatidylinositol metabolism[11]. Seventy per cent of chr16:70790626 carriers are from the Anogia village. The average relatedness (pairwise $\hat{\pi}$) is significantly higher in carriers (empirical $P = 0.006$ from 100,000 permutations), who are on average 11 years younger than non-carriers, $P = 6.00 \times 10^{-3}$ (Supplementary Table 6).

We identify a cardioprotective signal (rs145556679, EAF 0.013), which is associated with decreased triglycerides (TG) (beta $-1.13$ (SE 0.17), $P = 2.53 \times 10^{-11}$) and with very low-density lipoprotein cholesterol (VLDL) levels (beta $-1.13$

(SE 0.17), $P = 2.90 \times 10^{-11}$) (Table 2 and Fig. 5). This variant is not seen in any other worldwide cohort in the 1000 Genomes Project except for a single heterozygote reported in Toscani in Italia (TSI) samples ($n = 107$, MAF $= 0.005$) (Supplementary Table 7). However, as singletons were filtered out of the reference WGS data prior to phasing, rs145556679 is only represented in the MANOLIS sequences in the reference panel. Variants in LD with rs145556679 are present in haplotypes from other reference panel populations and a reduced signal for a different variant is detected when MANOLIS sequences are not included in the reference panel (Fig. 5e,f). This variant is the strongest new signal for TG/VLDL in MANOLIS and explains 3.21% and 3.20% of TG and VLDL variance, respectively. rs145556679 is located 942 kbp downstream of rs76353203 (*APOC3* R19X, previously associated with lipid traits[12,13], $r^2 = 0.001$). Conditional analysis confirms that rs145556679 is independent of R19X (TG, $P_{cond} = 1.09 \times 10^{-12}$; VLDL, $P_{cond} = 1.22 \times 10^{-12}$) (Supplementary Table 8). rs145556679 resides within an intron of the Down syndrome cell adhesion molecule like 1 (*DSCAML1*) gene, which is involved in cell adhesion in neuronal processes and is expressed in heart, liver, pancreas, skeletal muscle, kidney and brain[14,15]. An independent variant in this gene (rs10892151, 112 kbp away from rs145556679, $r^2 = 0.0005$ in MANOLIS) has previously been implicated with TG levels in the Amish founder population[13].

In MANOLIS we also observe an association between waist-to-hip ratio (WHR) and rs140087759 (beta 1.19 (SE 0.21), $P = 1.35 \times 10^{-8}$, EAF 0.010) (Table 2), located 5 kbp upstream

**Table 1 | Summary statistics at established loci.**

| Variant and cohorts | Trait | Chr:pos (EA/NEA) | Variant consequence | EAF | Beta (SE) | P-value | N | Reported variant | Reported genes | Reported PMID | Conditional P |
|---|---|---|---|---|---|---|---|---|---|---|---|
| rs7412 MANOLIS & Pomak | LDL | 9:45412079 (T/C) | Missense p.Arg176Cys | 0.079 | −0.419 (0.047) | $2.64 \times 10^{-19}$ | 3168 | rs7412 | APOC1, APOC2, APOE | 22286219 | NA |
| | TC | | | 0.079 | −0.27 (0.047) | $1.05 \times 10^{-8}$ | 3170 | | | | |
| rs7553007 MANOLIS & Pomak | CRP | 1:159698549 (A/G) | Intergenic | 0.327 | −0.202 (0.029) | $6.80 \times 10^{-12}$ | 2689 | rs7553007 | CRP | 19567438 | NA |
| rs964184 MANOLIS & Pomak | VLDL | 11:116648917 (G/C) | 3′ UTR | 0.163 | 0.242 (0.035) | $3.68 \times 10^{-12}$ | 3170 | rs964184 | APOA1 | 24097068 | NA |
| | TG | | | 0.163 | 0.236 (0.035) | $1.52 \times 10^{-11}$ | 3164 | | | | |
| rs76353203 MANOLIS | TG | 11:116701353 (T/C) | Stop-gain p.Arg19Ter | 0.022 | −1.073 (0.129) | $6.88 \times 10^{-17}$ | 1461 | rs76353203 | APOC3 | 24343240 | NA |
| | HDL | | | 0.022 | 0.919 (0.13) | $1.78 \times 10^{-12}$ | 1465 | | | | |
| rs150641967 MANOLIS & Pomak | LDL | 19:19370340 (T/TGACA) | Intronic | 0.075 | −0.326 (0.049) | $3.49 \times 10^{-11}$ | 3168 | rs10401969 | CILP2 | 24097068 | $9.34 \times 10^{-1}$ |
| | TC | | | 0.074 | −0.322 (0.046) | $8.29 \times 10^{-11}$ | 3170 | | | | $8.71 \times 10^{-1}$ |
| | TG | | | 0.074 | −0.278 (0.05) | $2.49 \times 10^{-8}$ | 3164 | | | | $3.94 \times 10^{-1}$ |
| | VLDL | | | 0.075 | −0.282 (0.05) | $1.48 \times 10^{-8}$ | 3170 | | | | $3.51 \times 10^{-1}$ |
| rs35237252 MANOLIS & Pomak | HDL | 8:19870271 (A/C) | Regulatory region | 0.277 | 0.183 (0.029) | $4.04 \times 10^{-10}$ | 3172 | rs2083637 | LPL | 19060911 | $1.39 \times 10^{-1}$ |
| rs200751500 MANOLIS & Pomak | HDL | 16:57001274 (A/AC) | Intronic | 0.33 | 0.294 (0.028) | $4.02 \times 10^{-25}$ | 3172 | rs1532624 | CETP | 19060911 | $1.18 \times 10^{-4}$ |
| rs1331309 MANOLIS & Pomak | MCH | 6:135406178 (G/T) | Intronic | 0.228 | 0.201 (0.033) | $1.90 \times 10^{-9}$ | 2829 | rs7775698 | MYB, HBS1L | 20139978 | $3.59 \times 10^{-1}$ |
| rs9804550 Pomak | WBC | 11:5186093 (T/C) | Intronic | 0.051 | 0.52 (0.081) | $1.10 \times 10^{-10}$ | 1673 | rs7116019 | TRIM68 | 25373335 | $5.33 \times 10^{-6}$ |
| | MCH | | | 0.053 | −0.627 (0.079) | $2.19 \times 10^{-15}$ | 1647 | | | | $1.43 \times 10^{-2}$ |
| | MCHC | | | 0.054 | 0.894 (0.075) | $8.46 \times 10^{-33}$ | 1669 | | | | $1.46 \times 10^{-4}$ |
| | MCV | | | 0.052 | −1.071 (0.076) | $1.57 \times 10^{-45}$ | 1658 | | | | $2.71 \times 10^{-5}$ |
| | RBC | | | 0.054 | 0.473 (0.077) | $8.58 \times 10^{-10}$ | 1718 | | | | $3.56 \times 10^{-2}$ |

Lead variants for validated, previously-reported association signals reaching $P < 5.00 \times 10^{-8}$. Cohorts, cohorts from which the signal arose; Chr:pos, represents the chromosome & position in GRCh37/hg19 coordinates; Variant consequence, taken from Ensembl (http://www.ensembl.org) the Human Genome Variation Society variant nomenclature (http://www.HGVS.org/varnomen) are provided for exonic variants. The other abbreviations are: EA, effect allele; NEA, non-effect allele; EAF, effect allele frequency; P, the Wald test P-value from the association analysis using METACARPA; N, sample size; Reported variant, RS-id of the reported signal; Reported genes, the gene(s) in which the signal was reported; reported PMID, PubMed ID for the reported GWAS signal; Conditional P, Wald test P from the association analysis using METACARPA of the variant after conditioning on the reported variant, confirming the signals are conditionally dependent; NA, indicates that conditional analysis is not applicable since the variant is the same as the reported variant; LDL, low-density lipoprotein cholesterol; TC, total cholesterol; CRP, C-reactive protein; VLDL, very low-density lipoprotein cholesterol; TG, triglycerides; HDL, high-density lipoprotein cholesterol; MCH, mean corpuscular haemoglobin; WBC, white blood cells; MCHC, mean corpuscular haemoglobin concentration; MCV, mean corpuscular volume; RBC, red blood cells.

of the long non-coding RNA gene *CTD-2061E9.1*. The signal is not associated with WHR in the Pomak population ($P = 0.39$), and has a higher frequency in the Pomak (MAF 0.038) and 1000 Genomes Project EUR populations (MAF 0.014) compared to MANOLIS (MAF 0.01). rs140087759 has no proxies with $r^2 > 0.8$ in MANOLIS and is not present in the WHR GWAS summary statistics from the Genetic Investigation of ANthropometric Traits (GIANT) study (https://www.broadinstitute.org/collaboration/giant/index.php/GIANT_consortium_data_files)[16].

In the Pomak population, we identify an association between diastolic blood pressure (DBP) and rs13382259 (beta 0.55 (SE 0.1), $P = 3.18 \times 10^{-8}$, EAF 0.043) (Table 2), which resides in a predicted promoter (*ENSR00000596922*)[17] in an intron of *PSD4*. It is located 1 kbp upstream of novel transcript *AC016683.5*. rs13382259 is associated with the expression of *PAX8* in tibial nerve (GTEx Portal, http://www.gtexportal.org). The allele frequency of rs13382259 is lower in the MANOLIS (MAF 0.024) compared with the 1000 Genomes Project EUR populations (MAF 0.05) and the Pomak population (MAF 0.05). The signal is not associated in MANOLIS ($P = 0.53$) and is not present in the genome-wide summary statistics for the International Consortium for Blood Pressure (ICBP)[18]. Proxies for rs13382259 ($r^2 > 0.8$) are present in the International HapMap Project data (http://hapmap.ncbi.nlm.nih.gov) and three are present in ICBP summary statistics but none were significantly associated with DBP.

We also identify an association between fasting glucose levels adjusted for BMI (FGBMIadj) and rs6131100 (beta −0.79 (SE 0.14), $P = 1.21 \times 10^{-8}$, EAF 0.037) (Table 2). rs6131100 is situated in the intron of *SLX4IPA* and 20 kbp upstream of *MKKS*, which is associated with Bardet–Biedl syndrome 6 (OMIM: 605552). The allele frequency of rs6131100 is higher in the MANOLIS (MAF 0.083) and 1000 Genomes Project EUR populations (MAF 0.053) compared to the Pomak population (MAF 0.039). rs6131100 is not associated with FGBMIadj in MANOLIS ($P = 0.91$), and is not present in genome-wide summary data available from the Meta-Analyses of Glucose and Insulin-related traits Consortium (MAGIC) study (www.magicinvestigators.org)[19–21]. One proxy for rs6131100 was present in the International HapMap Project but this did not show evidence of association in the MAGIC genome-wide summary data for FGBMIadj.

In the Pomak cohort, we also observe an association with white blood cell count (WBC) and rs79748197 (beta −1.16 (SE 0.21), $P = 3.00 \times 10^{-8}$, EAF 0.008) (Table 2), which resides in the intron of a non-coding transcript (*AC092594.1*). The closest protein-coding gene is *OSR1*, 121 kbp away, a widely-expressed transcription factor implicated in embryonic heart, kidney and urogenital development[22]. rs79748197 has a similar frequency in MANOLIS and is not associated with WBC ($P = 0.19$). It has a higher allele frequency in the 1000 Genomes Project EUR population (MAF 0.014). No proxies are present for rs79748197

**Table 2 | Summary of novel association signals.**

| Variant and cohorts | Trait | Chr:pos (EA/NEA) | Nearest gene | Internal replication | | | | | EAF | Beta (SE) | P-value | Overall MAC (N) |
|---|---|---|---|---|---|---|---|---|---|---|---|---|
| | | | | Replication cohorts | EAF | Beta (SE) | P-value | MAC (N) | | | | |
| chr16:70790626 MANOLIS | HDL | 16:70790626 (T/C) | VAC14-AS1 VAC14 | MANOLIS CoreExome | 0.003 | −1.885 (0.994) | $5.76 \times 10^{-2}$ | 1.26 (210) | 0.006 | −1.713 (0.254) | $1.57 \times 10^{-11}$ | 20 (1476) |
| | | | | MANOLIS OmniExome | 0.007 | −1.702 (0.263) | $1.81 \times 10^{-10}$ | 17.6 (1255) | | | | |
| rs145556679 MANOLIS | TG | 11:117643264 (C/G) | DSCAML1 | MANOLIS CoreExome | 0.005 | −1.293 (0.729) | $7.85 \times 10^{-2}$ | 2.09 (209) | 0.013 | −1.134 (0.17) | $2.53 \times 10^{-11}$ | 49 (1476) |
| | | | | MANOLIS OmniExome | 0.014 | −1.125 (0.175) | $1.70 \times 10^{-10}$ | 35.1 (1252) | | | | |
| | VLDL | | | MANOLIS CoreExome | 0.005 | −1.365 (0.727) | $6.21 \times 10^{-2}$ | 2.1 (210) | 0.013 | −1.131 (0.17) | $2.90 \times 10^{-11}$ | |
| | | | | MANOLIS OmniExome | 0.014 | −1.118 (0.175) | $2.29 \times 10^{-10}$ | 35.1 (1253) | | | | |
| rs140087759 MANOLIS | WHR | 5:28292892 (T/C) | CTD-2061E9.1 | MANOLIS CoreExome | 0.015 | 1.676 (0.411) | $5.92 \times 10^{-5}$ | 6.12 (204) | 0.01 | 1.189 (0.209) | $1.35 \times 10^{-8}$ | 31 (1476) |
| | | | | MANOLIS OmniExome | 0.009 | 1.02 (0.243) | $2.90 \times 10^{-5}$ | 18.8 (1047) | | | | |
| rs13382259* Pomak | DBP | 2:113934176 (T/A) | PSD4 | Pomak CoreExome | 0.047 | 0.509 (0.126) | $6.98 \times 10^{-5}$ | 60.3 (641) | 0.043 | 0.554 (0.1) | $3.18 \times 10^{-8}$ | 172 (1737) |
| | | | | Pomak OmniExome | 0.039 | 0.629 (0.164) | $1.36 \times 10^{-4}$ | 43 (551) | | | | |
| rs6131100* Pomak | FGBMIadj | 20:10434530 (A/T) | SLX4IP | Pomak CoreExome | 0.038 | −0.573 (0.16) | $3.62 \times 10^{-4}$ | 43.2 (569) | 0.037 | −0.79 (0.139) | $1.21 \times 10^{-8}$ | 135 (1737) |
| | | | | Pomak OmniExome | 0.035 | −1.454 (0.279) | $7.12 \times 10^{-7}$ | 12.2 (174) | | | | |
| rs79748197 Pomak | WBC | 2:19430105 (G/A) | AC092594.1 | Pomak CoreExome | 0.004 | −1.242 (0.403) | $2.12 \times 10^{-3}$ | 5.8 (725) | 0.008 | −1.156 (0.209) | $3.00 \times 10^{-8}$ | 31 (1737) |
| | | | | Pomak OmniExome | 0.004 | −1.125 (0.243) | $4.14 \times 10^{-6}$ | 20.9 (948) | | | | |
| rs557129696 Pomak | HGB | 11:5328683 (G/T) | HBG2 HBE1 AC104389.28 | Pomak CoreExome | 0.002 | −1.95 (0.606) | $1.36 \times 10^{-3}$ | 2.87 (717) | 0.004 | −2.027 (0.308) | $4.83 \times 10^{-11}$ | 13 (1737) |
| | | | | Pomak OmniExome | 0.005 | −2.054 (0.358) | $1.30 \times 10^{-8}$ | 9.45 (945) | | | | |
| rs112037309* MANOLIS & Pomak | Weight | 4:106617136 (A/G) | ARHGEF38 INTS12 | MANOLIS | 0.075 | 0.295 (0.078) | $1.43 \times 10^{-4}$ | 189.8 (1258) | 0.075 | 0.287 (0.052) | $2.70 \times 10^{-8}$ | 485 (3213) |
| | | | | Pomak | 0.075 | 0.28 (0.07) | $5.96 \times 10^{-5}$ | 250.8 (1672) | | | | |

All variants are intronic with the exception of rs140087759 which is intergenic, variant consequences are taken from Ensembl (http://www.ensembl.org). For the internal replication the software used was GEMMA with the exception of rs112037309 in which METACARPA was used. Cohorts, cohorts from which the signal arose. Chr:pos, represents the chromosome and position in GRCh37/hg19 coordinates; EA, effect allele; NEA, non-effect allele; EAF, effect allele frequency; P-value, the likelihood ratio test P-value from GEMMA or Wald test P-value from METACARPA; MAC, minor allele count for samples in the analysis; Overall MAC, minor allele count for all samples in the cohorts from which the signal arose, established using the rounded imputed allele dosages from SNPTEST (https://mathgen.stats.ox.ac.uk/genetics_software/snptest/snptest.html); N, sample size; HDL, high-density lipoprotein cholesterol; DBP, diastolic blood pressure; TG, triglycerides; VLDL, very low-density lipoprotein cholesterol; FGBMIadjusted, fasting glucose adjusted for body mass index; HGB, haemoglobin; WBC, white blood cells; WHR, waist-to-hip ratio.
*At least one proxy is present in the International HapMap project data (http://hapmap.ncbi.nlm.nih.gov). Proxies were determined using LD ($r^2 > 0.8$ in the cohorts used for the meta-analysis) for each novel variant. If a proxy was in HapMap it also had high LD ($r^2 > 0.9$) with the variant in the 1000 Genomes Project CEU population[3]. LocusZoom was used to create the regional plots (http://csg.sph.umich.edu/locuszoom/).

in the Pomak population and this trait was not examined in the Haemgen RBC study[23].

Lastly, rs557129696 is associated with haemoglobin levels (HGB) (beta − 2.03 (SE 0.31), $P = 4.83 \times 10^{-11}$, EAF 0.004) (Table 2). The variant resides in an extended LD region spanning 1.4 Mb. The signal is conditionally independent (Supplementary Table 8) of previously-reported blood trait GWAS signals in this region in the same population[24]. rs557129696 resides in the intronic regions of two haemoglobin-coding genes (MBE1 and MBG1) and a non-coding RNA gene (AC104389.28). The G-allele of rs557129696 is not seen in the 1000 Genomes EUR population. Numerous associations with red blood cell traits, anaemia and thalassemias have been linked to this chromosome 11 region[25–28]. We have previously observed an independent signal associated with blood traits in this chromosomal region of extended LD[24]. Notably, associations between variants in this region and foetal haemoglobin levels[29] have been reported in the Sardinian founder population.

Weight was associated with rs112037309 (beta 0.29 (SE 0.05), $P = 2.70 \times 10^{-8}$, EAF 0.075) in both isolated populations (Table 2). rs112037309 is situated in the intronic regions of ARHGEF38 and INTS12. The protein product of ARHGEF38 is involved in signal transduction, and immunohistochemistry shows strong staining in pancreatic islets, skeletal and smooth muscle (Protein atlas, http://www.proteinatlas.org). rs112037309 has a higher frequency in the 1000 Genomes Project EUR population (MAF 0.096) compared with the Pomak (MAF 0.073)

and MANOLIS (MAF 0.074) populations. We were unable to look up this variant in large GWAS studies as weight is not one of the traits included as part of the Genetic Investigation of Anthropometric Traits (GIANT)[16] study.

## Discussion

We provide here a first characterization of the genetic architecture of the MANOLIS isolated population and report 9.5M SNVs, of which 6% are absent from previous sequenced panels and are enriched for predicted functional consequences. Our complex trait association findings highlight the advantages of whole genome sequencing in founder populations: two lipid traits and the HGB signals we identify are driven by variants unique to the MANOLIS cohort or extremely rare in other worldwide populations. The remaining five novel associations are present in European populations (1000 Genomes Project EUR MAF ranging from 0.014 to 0.096) but are not significantly associated in GWAS meta-analyses of cosmopolitan populations. This can be due to a number of reasons in addition to winner's curse, that is, larger effect sizes in the discovery isolate cohort. For two of these signals, the variant and its proxies are not present in the HapMap reference panel and therefore these variants are not represented in GWAS conducted to date. Three of the associated variants are represented in HapMap and show no evidence of association outside the isolate; this can indicate that the

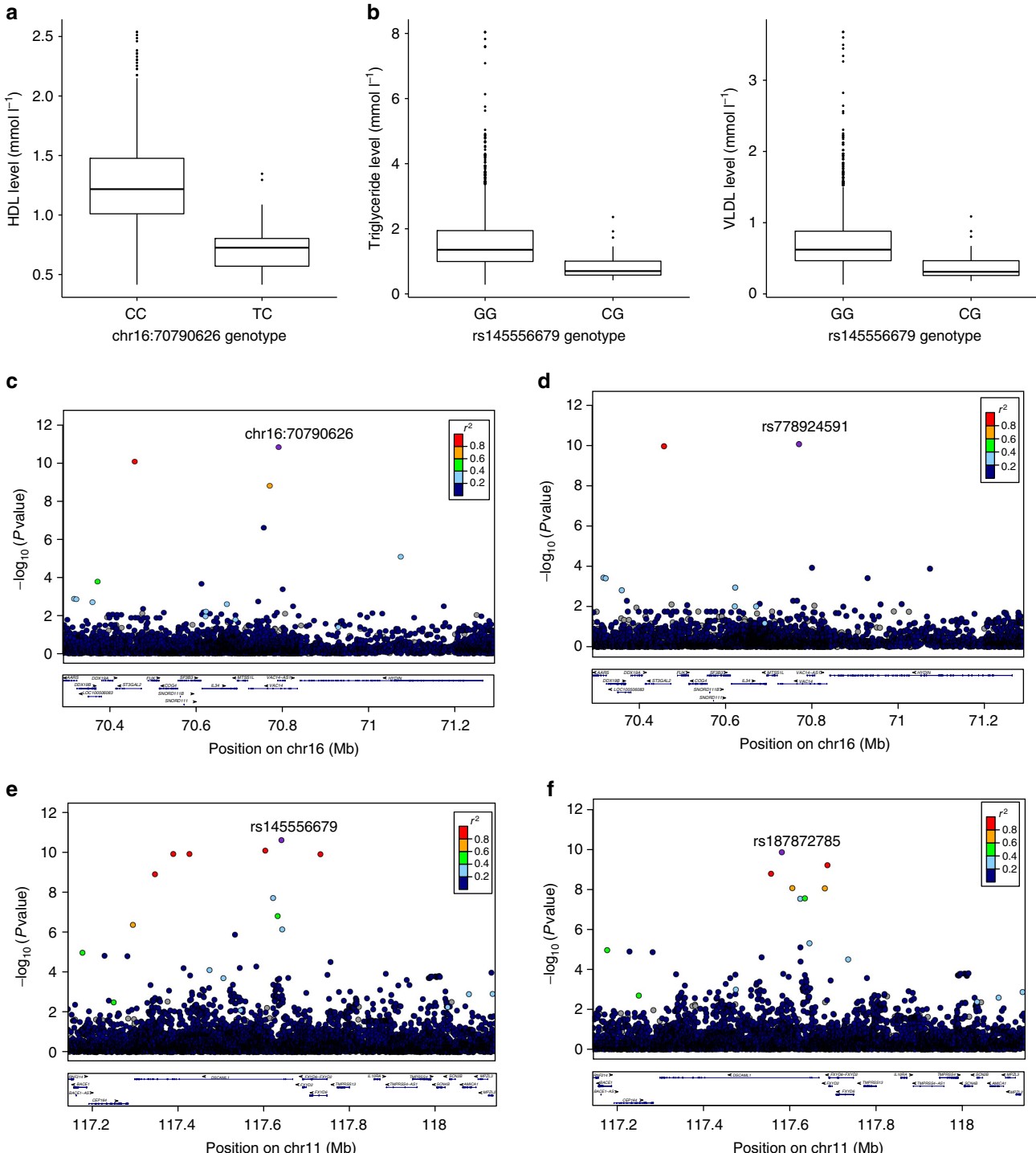

**Figure 5 | Association results for chr16:70790626 and rs145556679 and lipid levels. (a)** Heterozygotes for chr16:70790626 exhibit significantly lower HDL levels than homozygotes (Wald test METACARPA $P = 1.57 \times 10^{-11}$). **(b)** Heterozygotes for rs145556679 exhibit significantly lower TG (Wald test METACARPA $P = 2.53 \times 10^{-11}$) and VLDL (Wald test METACARPA $P = 2.90 \times 10^{-11}$) levels than homozygotes. **(c)** Regional association plot for chr16:70790626. **(d)** To determine if the signals are detected without MANOLIS sequences in the reference panel, we conducted imputation using a combined UK10K + 1000 Genomes reference panel; the regional plot shows that the chr16:70790626 signal is captured with a different lead variant and a decrease in significance. **(e)** Regional association plot for rs145556679. **(f)** Regional association plot for rs145556679 using a combined UK10K + 1000 Genomes reference panel; the same signal is captured with a different lead variant and a decrease in association strength. LocusZoom was used to create the regional plots (http://csg.sph.umich.edu/locuszoom/).

index variant is in LD with the causal variant in the isolate but not in the cosmopolitan population. Furthermore, the effect and therefore the power to detect associations can be increased in isolates due to the environmental and phenotypic homogeneity when compared to other worldwide populations, in addition to extended LD.

Our study demonstrates the power benefits of using a large, sequence-based imputation reference panel. Six of the eight new

associations reported here have been detected for the first time as a consequence of improved imputation accuracy. The cost-effective hybrid WGS and imputation approach in founder populations serves as a good model for further low-frequency variant discovery, which can enhance our understanding of the biological processes underpinning complex traits of medical relevance.

## Methods

**Cohort descriptives.** The HELIC (Hellenic Isolated Cohorts; www.helic.org) MANOLIS (Minoan Isolates) collection focuses on Anogia and surrounding Mylopotamos villages on the Greek island of Crete. All individuals had to have at least one parent from the Mylopotamos area. The HELIC Pomak collection focuses on the Pomak villages, a set of isolated mountainous villages in the North of Greece. Recruitment of both population-based samples was primarily carried out at the village medical centres. The study includes biological sample collection for DNA extraction and lab-based blood measurements, and interview-based questionnaire filling. The phenotypes collected include anthropometric and biometric measurements, clinical evaluation data, biochemical and haematological profiles, self-reported medical history, demographic, socioeconomic and lifestyle information. The study was approved by the Harokopio University Bioethics Committee and informed consent was obtained from every participant.

**HELIC MANOLIS sequencing data.** *Sample selection.* Overall, 250 HELIC MANOLIS samples were whole-genome sequenced at $4\times$ depth to provide reference haplotypes for imputation. To maximize haplotype diversity, the 250 most unrelated samples were selected from a set of 1,118 samples genotyped on the Illumina OmniExpress chip. Common (MAF>5%) variants were used to calculate the pairwise identity by descent; there were 624,403 such pairs for 1,118 individuals. Samples were selected based on a maximal $\hat{\pi}$ of 0.15 across all samples.

*Sequencing and variant calling.* Sequencing was performed at $4\times$ average depth using Illumina HiSeq 2000 sequencers. The data was aligned to the 1000 Genomes Project phase 1 reference assembly using BWA[30]. Optical and PCR duplicates were removed using Picard MarkDuplicates (http://broadinstitute.github.io/picard). Variants were called using samtools[31] mpileup, and quality score recalibration was performed using the variant quality score recalibration (VQSR) tool from the GATK[32] v.2.1.13 suite. After recalibration, one sample was found to be an ethnic outlier and was removed. No samples were excluded based on concordance checks with genotype data, sex checks, mean depth per sample, heterozygous or singleton rate per sample or non-reference allele (NREF) discordance.

*Variant QC and Haplotype creation.* Post-VQSR, variants were filtered so as to yield a sensitivity threshold of 90% for INDELS (VQSLOD<3.1159) and a threshold of 94% for SNPs (VQSLOD<5.4079).Variants were excluded if they were multi-allelic, monomorphic, singletons, indels, had a missingness >3% or a HWE $P<1.00\times10^{-4}$. Any variant from the HELIC MANOLIS data for which the alleles differed from the 1000 Genomes Project and UK10K datasets at the same position was excluded. Phasing was performed using SHAPEIT v2.r727 (ref. 33). Following imputation, variants with IMPUTEv2 (ref. 34) info score <0.7 were filtered out. The final imputed variant set shows excellent genotype and minor allele concordance across the MAF spectrum compared to the array data (Supplementary Fig. 5). Average minor allele concordance was 94.6% for rare (MAF<1%) variants, 96.7% for low-frequency (1%<MAF<5%) variants and 99.6% for common variants (MAF>5%). SNP density inside low-complexity regions (LCR) in the hg19 build was 6.5 times lower than in the accessible genome (Supplementary Note 2 and Supplementary Table 9).

**Merged reference panel creation.** A large reference panel was constructed by combining the WGS haplotypes from HELIC MANOLIS (9,554,503 variants and 249 samples), 1000 Genomes Project[35] (27,449,245 variants and 1092 samples), and UK10K[1] (25,109,897 variants and 3781 samples). For 1000 Genomes Project we used 1,000 Genomes Project haplotypes Phase I integrated variant set release (SHAPEIT2) in NCBI build 37 (hg19) coordinates. All ethnicities with singletons excluded (ALL.integrated_phase1_SHAPEIT_16-06-14.nosing) downloaded from the IMPUTEv2 (refs 33,34) website (http://mathgen.stats.ox.ac.uk/impute/impute_v2.1.0.html). For UK10K the haplotypes were prepared and described previously[1,2].

IMPUTEv2 (refs 33,34) was used to merge the haplotypes in a two-step process; firstly merging the 1000 Genomes Project with the UK10K datasets and secondly merging the HELIC MANOLIS with the UK10K-1000 Genomes Project reference haplotypes. The merged reference panel contained 5,211 samples and 38,810,554 variants.

**HELIC MANOLIS WGS SNV frequencies and functional annotation.** The human genome was split to functional regions as follows: coding and UTR sequences were extracted from GENCODE annotations (Release 19, mapped to GRCh37 build[36]), upstream/downstream regions, introns and splice regions were derived from GENCODE data following Sequence Ontology definitions[37]. Intergenic regions were defined as regions where no GENCODE genes were

overlapping the ungapped human genome (Ensembl release 75, build GRCh37). Bedtools[38] was used to find overlapping variants with each genomic regions then the average frequency was calculated (Fig. 2a).

Using Ensembl variant effect predictor (VEP) (http://www.ensembl.org, version 75, on build GRCH37)[39], the most severe consequence term was assigned to each autosomal SNV in the phased and imputed dataset (the same dataset is used in all subsequent analyses). Consequences were pooled into eight consequence categories: 3′/5′ UTR, coding sequence variant, intergenic variant, intron variant, splice-region variant, non-coding transcript variant, upstream/downstream variant and regulatory variant (Supplementary Table 10). Variants were grouped into the following bins: MAC=2, MAC>2 and MAF≤1%, 1%<MAF≤2%, 2%<MAF≤5%, MAF>5%. The percentage of variants with a given consequence term in each MAF bin was calculated (Supplementary Table 2 and Supplementary Fig. 1).

**HELIC MANOLIS $4\times$ WGS reference panel comparisons.** The autosomal SNVs in the 249 HELIC MANOLIS $4\times$ WGS reference haplotype dataset were binned according to the observed MAF. Each variant was checked to establish if it was present in the UK10K ($n=3718$) and/or the 1000 Genomes Project ($n=1092$) reference haplotype dataset (Fig. 2).

**Functional enrichment of variants private to HELIC MANOLIS.** Variants discovered in the $4\times$ WGS were separated into two groups, those variants shared with UK10K or the 1000 Genomes reference dataset (9,030,004 variants) or those unique to HELIC MANOLIS (524,499 variants). To compare the distributions of consequence terms in the shared and the unique datasets a two-sided proportion test was used for each MAF bin, using the consequence and MAF bins described above. Fold enrichment was calculated with the following equation:

$$E^{M,c} = \frac{C_u^{M,c}/t_u^M}{C_s^{M,c}/t_s^M}.$$

Where $E^{M,c}$ is the fold enrichment of consequence $c$ in $M$ MAF bin; $C_u^{M,c}$ and $C_s^{M,c}$ are the number of variants in $M$ MAF bin with consequence $c$ in the unique and the shared dataset respectively; $t_u^M$ and $t_s^M$ are the total number of variants in the $M$ MAF bin in the unique dataset and the shared dataset respectively. Bonferroni correction of the $P$ was applied to account for multiple testing (Fig. 3 and Supplementary Table 4).

**Array genotyping and quality control.** The MANOLIS and Pomak cohorts were each genotyped in two tranches: one on the Illumina HumanOmniExpress BeadChip and Illumina HumanExome BeadChip, and one on the Illumina HumanCoreExome beadchip (Illumina, San Diego, CA, USA) at the Wellcome Trust Sanger Institute, Hinxton, UK. The two datasets for each cohort were phased, imputed and analysed separately (Fig. 1).

Quality control (QC) for the samples genotyped using the OmniExpress genotypes has been previously described[24]. The same samples were genotyped using the HumanExome BeadChipv1.1 at the Wellcome Trust Sanger Institute, Hinxton, UK and called with Illumina Genome Studio Gencall, and zCall[40]. The calling and QC were undertaken separately for the Pomak and MANOLIS cohorts using a step-wise QC approach which consisted of GenCall sample QC followed by zCall sample and variant QC (Supplementary Methods). The genotypes from the OmniExpress and HumanExome chips were merged into a single dataset. For variants present in OmniExpress and HumanExome the genotypes for those with MAF≥5% were taken from the OmniExpress while those with MAF<5% were taken from the HumanExome. This merged genotype, referred to as the 'OmniExome' dataset, contained 1265 samples and 621,908 variants for the MANOLIS and 1003 samples and 612,403 variants for Pomak.

Additional HELIC MANOLIS and Pomak samples were genotyped on the Illumina HumanCoreExome-12-v1.1 (Illumina) at the Wellcome Trust Sanger Institute, Hinxton, UK. Genotypes were called with GenCall and zCall (Supplementary Methods). In MANOLIS 211 samples and 529,604 variants, and in Pomak 734 samples and 529,086 variants passed QC.

**Phasing and imputation.** Each cohort and array was phased and imputed separately (Fig. 1). Before phasing, variants were excluded that were duplicates, monomorphics, singleton variants, had poor intensity clustering, or had allelic differences between the array and reference panel. Samples not genotyped on both the OmniExpress and Exome chip arrays were excluded, as well as variants with MAF <5% genotyped on the OmniExpress. Samples were phased using SHAPEIT v2.r778 (ref. 33) and imputed using IMPUTE v2.3.1 (refs 33,34). Following imputation, any variant with HWE $P<1.00\times10^{-4}$ or imputation information score <0.4 was excluded. There was good genotype concordance between the 249 overlapping samples in the imputed and WGS (Supplementary Note 3).

**Phenotype preparation.** Thirty-one phenotypes encompassing cardiometabolic, anthropomorphic and haematological traits were prepared separately for each cohort and array (Supplementary Table 11). If gender differences were significant (Wilcoxon rank sum, $P<0.05$), the phenotype was stratified accordingly. Following trait-specific

exclusions and adjustments, outliers were filtered out based on 3, 4 or 5 SD away from the mean. Traits not normally distributed were transformed to normality using either an inverse normal or log transformation. For all traits age and $age^2$ were added as covariates as necessary and standardised residuals were used. Some traits are adjusted for body mass index (BMI). If male and female phenotypes were prepared separately these were standardised before combining the residuals. Summary statistics for all of the traits are provided in Supplementary Table 12.

**Association analysis.** *GEMMA.* Association analysis was performed separately for each cohort and array using the imputed genotypes. GEMMA[41] was used for the analysis. This software allows accounting for relatedness at the array level by using a linear mixed model. A centred kinship matrix was generated using only the directly typed array genotypes. $P$ values from the likelihood ratio test (p_lrt) are reported. For meta-analysis within and across cohorts we developed and used METACARPA.

*METACARPA.* When meta-analysing GWAS, both the $P$ values and effect sizes can be meta-analysed on a per-variant basis. Both meta-statistics are weighted sums., for example, for effect sizes:

$$\hat{\eta} = \sum_{k=1}^{K} w_k \hat{\eta_k},$$

where $\hat{\eta}$ is the estimator of a common effect $\eta$ across all studies, $k \in \{1...K\}$ identifies the study among the $K$ that should be meta-analysed, $\hat{\eta_k}$ is the effect in study $k$ and $w_k$ is a study-specific weight. For $P$ values, we transform to $z$-scores using $z_k = \Phi^{-1}\left(P_k/2\right) \times \text{sgn}(\eta_k)$, where $\Phi$ is the cumulative distribution of the standard normal. Then:

$$\hat{z} = \sum_{k=1}^{K} w_k z_k.$$

Then, $z$-scores are transformed back to $P$ with the complement of the previous transformation: $p_{\text{meta}} = 2\Phi_{0,\sigma}(-|\hat{z}|)$. In both cases, the variance $\sigma \neq 1$ needs to be derived. For both $\hat{z}$ and $\hat{\eta}$ it has the typical form of a variance of weighted sums:

$$\text{Var}(\hat{\eta}) = \sum_{k=1}^{K} w_k^2 \text{Var}(\hat{\eta_k}) + 2 \sum_{k=1}^{K} \sum_{l=k+1}^{K} w_k w_l \text{Cov}(\hat{\eta_k}, \hat{\eta_l}),$$

$$\text{Var}(\hat{z}) = \sum_{k=1}^{K} w_k^2 \text{Var}(z_k) + 2 \sum_{k=1}^{K} \sum_{l=k+1}^{K} w_k w_l \text{Cov}(z_k, z_l).$$

$\text{Var}(z_k) = 1$ by construction and $\text{Var}(\hat{\eta_k})$ is taken from the input files. The previous equations require the covariances of the individual study statistics across all pairs of studies. We build a $K \times K$ variance-covariance matrix describing this 'inter-study relatedness'. Lin and Sullivan[8] propose the following for estimating study correlation in quantitative trait GWAS:

$$\text{Corr}(\hat{\eta_k}, \hat{\eta_l}) \approx \frac{n_{kl}}{\sqrt{n_k n_l}},$$

which is the number of overlapping individuals $n_{kl}$ in relation to the studies sample sizes $n_k$ and $n_l$. However, in many cases $n_{kl}$ is unknown, or the relatedness is subtler than a simple overlap. Province and Borecki[7] propose the following:

$$\text{Corr}(z_k, z_l) \approx r_{\text{tetrachoric}}(z_{k_{0|1}}, z_{l_{0|1}}),$$

where $z_{k_{0|1}} = \begin{cases} 1 \text{ if } z_k \geq 0 \\ 0 \text{ if } z_k < 0 \end{cases}$, and $r_{\text{tetrachoric}}$ is the tetrachoric correlation coefficient. We obtain covariances using $\text{Cov}(x, y) = \sigma_{xy} = \sigma_x \sigma_y r_{xy}$, since $\sigma_x$ and $\sigma_y$ the variances in each study, are known. It is assumed that for every $(k,l)$, $r_{k,l} = \text{Corr}(z_k, z_l) = \text{Corr}(\hat{\eta_k}, \hat{\eta_l})$, that is, the general term for the variance-covariance matrix $\Omega_z$ for the $P$ meta-analysis is $\omega_{z_{k,l}} = r_{k,l}$, and the general term for the variance-covariance matrix $\Omega_\eta$ for the effect-size meta-analysis is $\omega_\eta k, l = r_{k,l} \sigma_k \sigma_l$.

For weights, it is shown[8] that in the case of overlapping samples, the $w_k$ are of the form:

$$\mathbf{w} = \frac{1}{\mathbb{1}^T \Omega_\eta^{-1} \mathbb{1}} \times \mathbb{1}^T \Omega_\eta^{-1},$$

where $\mathbb{1}$ is the unity vector of size $K$ and $\Omega_\eta$ is the estimated covariance matrix of the effect sizes between studies with general term $\omega_{\eta_{k,l}}$.

For the $P$ meta-analysis, the general term of $\Omega_z$ does not contain a factor accounting for unequal sample size. The following weight vector does:

$$\mathbf{w} = \frac{1}{\mathbb{1}^T \mathbf{s}} \times \mathbf{s},$$

where $s$ is a vector containing the sample sizes of all studies. The general term of the weight vector is $w_i = \frac{s_i}{\sum_k s_k}$, the relative sample size of study $i$.

We implemented this method in C++ using the Boost libraries. For tetrachoric correlation, we use the approximation of Digby[42], which has been shown to be valid when analysing equilibrated $2 \times 2$ tables of large sample sizes, which is the case when binary-transforming GWAS $P$ values. This result was confirmed by comparing the approximated value with an iterative maximum likelihood estimator.

*Simulation and benchmark.* This implementation was tested by repeatedly drawing two random sets of 2,000 samples each from the UKHLS GWAS dataset (EGA accession EGAD00001000890), with increasing sample overlap. Phenotypes were drawn from a standard normal. The two studies were associated separately using GEMMA[41], then meta-analysed using METACARPA, and the whole process was repeated 1,000 times for each level of overlap. An uncorrected fixed-effects, sample size-weighted $P$ value-based meta-analysis[43] was implemented in the software for comparison, as well as an uncorrected inverse-variance weighted, effect size-based meta-analysis. We used degrees of overlap ranging from 0.5 to 75% of the total sample size (Fig. 4 and Supplementary Fig. 2). We assessed the false-positive rate calculated at a genome-wide significance threshold of $5.00 \times 10^{-8}$, and the power to detect a single associated SNP. Effect SNPs were chosen randomly for each simulation, MAF and effect sizes were constrained so that the effect SNP explained 1% of phenotype variance.

For typical to substantial levels of overlap (0.5–10%), false-positive rate grows linearly for both the two uncorrected and the two corrected methods (Fig. 4a). However, for the latter, the growth rate is reduced from $6 \times 10^{-5}$%/sample to $5.5 \times 10^{-5}$%/sample (8.3%). While for typical (0.5–5%) levels of overlap, power to detect a single SNP is conserved, for substantial levels of overlap (5–10%) it drops at an approximate rate of 0.05%/sample. For extensive levels of overlap (10–75%), the increase in false-positive rate slows further and stabilizes around 9% for overlaps greater than 50% for both corrected methods (Supplementary Fig. 2), whereas uncorrected methods keep growing at an unchanged rate. Owing to the reduction in effective sample size, power decreases to below 60% for very high levels of overlaps. At the levels of overlap inferred in the HELIC datasets (1.96 and 1.84%), power is decreased by 0.1% and false-positive rate is decreased by 0.2% between the corrected and uncorrected effect-size based meta-analyses.

We evaluated the accuracy of tetrachoric correlation in estimating the true simulated sample overlap (Supplementary Fig. 6) compared to Pearson's correlation of $z$-scores. Although both methods systematically underestimated sample overlap, tetrachoric correlation performed poorly compared to Pearson's when all SNPs were under the null (Supplementary Fig. 6). Tetrachoric correlation's main advantage is to ignore outliers, hence it may be overconservative under the null. Under a simulated polygenic burden across 10,000 SNPs for a trait that is 20% heritable under 25% sample overlap, both methods overestimated correlation but tetrachoric correlation was more accurate than Pearson's (Fig. 4c). This suggests that tetrachoric correlation is able to correct for the presence of a relatively high number of truly associated, correlated SNPs, a scenario which is expected to arise when analysing highly polygenic traits.

*Implementation.* This method is implemented in the METACARPA software (META-analysis in C++ Accounting for Relatedness using arbitrary Precision Arithmetic). Binary and sources are freely available (https://github.com/wtsi-team144/metacarpa, http://www.sanger.ac.uk/science/tools/metacarpa).

**Prioritization and validation.** Variants were prioritized for validation by direct genotyping from the meta-analysis across cohorts, keeping only the most significant SNV with $P \leq 5.00 \times 10^{-8}$ in a 500 kbp window around any given signal. Variants that were genome-wide significant in the within-cohort meta-analysis and not within 500 kbp of the across cohort meta-analysis signal were also considered. Replication is demonstrated in the within-cohort meta-analysis by nominal significance (two-sided $P \leq 0.05$) in the same direction in both datasets. We relaxed this to $P \leq 0.08$ for MANOLIS CoreExome for 2 variants due to small sample size. For the across cohort meta-analysis both within-cohort meta-analysis have $P \leq 0.05$. To determine the Bonferroni corrected genome-wide significance level, for each cohort array we used the eigenvalues of the correlation matrix of the 31 traits tested[44] to calculate the effective number of independent phenotypes. Then, the genome-wide $P$ threshold to control FWER at 5%, using a Bonferroni correction, is $5.00 \times 10^{-8}$/effective number of independent phenotypes. We selected the cohort array with the maximum number of independent phenotypes for the calculation, which was $5.00 \times 10^{-8}/14.99 = 3.33 \times 10^{-9}$.

Prioritized variants were independently genotyped in as many of the imputed samples as possible using the Sequenom iPLEX Assay and the Sequenom MassARRAY System (Agena Bioscience) (Supplementary Methods).

Sixty two variants were directly genotyped in a maximum of 2,778 samples. Concordances of the major and minor alleles were calculated separately. The minor allele concordance and the positive predictive value (PPV), which is the fraction of true positives for the minor allele calls, were used to assess the imputed genotype quality. Phenotypes were prepared again only for the samples with directly typed genotypes and the association and meta-analysis were repeated.

Concordance and PPV were calculated as follows:

$$r = \frac{1}{n} \sum_{k=1}^{n} \frac{\text{No. concordant minor alleles}}{\text{No. minor alleles in reference GWAS data}},$$

$$\text{PPV} = \frac{1}{n} \sum_{k=1}^{n} \frac{\text{No. concordant minor alleles}}{\text{No. minor alleles in sequencing}}.$$

The proportion of variants that had both concordance and PPV > 90% were:

MANOLIS CoreExome 54.8%; MANOLIS OmniExome 53.2%; Pomak CoreExome 50%; Pomak OmniExome 54.8%. All variants reported here pass validation (Supplementary Table 5).

For the weight signal with rs112037309 we validated a proxy rs17262443 which has $r^2 = 1$ with rs112037309 (rs17262443, $P = 3.69 \times 10^{-8}$). Conditional analysis confirmed these represent the same signal.

**Data availability.** The following HELIC genotype and WGS datasets have been deposited to the European Genome-phenome Archive (https://www.ebi.ac.uk/ega/home): EGAD00010000518; EGAD00010000522; EGAD00010000610; EGAD00001001636. We have also contributed the 249 HELIC MANOLIS whole-genome sequences to the Haplotype Reference Consortium (http://www.haplotype-reference-consortium.org).

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

## Acknowledgements

This work was funded by the Wellcome Trust (WT098051) and the European Research Council (ERC-2011-StG 280559-SEPI). The MANOLIS study is dedicated to the memory of Manolis Giannakakis, 1978-2010. We thank the residents of the Pomak villages and of the Mylopotamos villages for taking part. The HELIC study has been supported by many individuals who have contributed to sample collection (including Antonis Athanasiadis, Olina Balafouti, Christina Batzaki, Georgios Daskalakis, Eleni Emmanouil, Pounar Feritoglou, Chrisoula Giannakaki, Margarita Giannakopoulou, Kiki Kaldaridou, Anastasia Kaparou, Vasiliki Kariakli, Stella Koinaki, Dimitra Kokori, Maria Konidari, Hara Koundouraki, Dimitris Koutoukidis, Vasiliki Mamakou, Eirini Mamalaki, Eirini Mpamiaki, Nilden Selim, Nesse Souloglou, Maria Tsoukana, Dimitra Tzakou, Katerina Vosdogianni, Niovi Xenaki, Eleni Zengini), data entry (Thanos Antonos, Dimitra Papagrigoriou, Betty Spiliopoulou), sample logistics (Sarah Edkins, Emma Grey), genotyping (Suzannah Bumpstead, Robert Andrews, Hannah Blackburn, Doug Simpkin, Siobhan Whitehead), research administration (Anja Kolb-Kokocinski, Carol Smee, Danielle Walker) and informatics (Kathleen Stirrups, Martin Pollard, Josh Randall). The GATK programme was made available through the generosity of Medical and Population Genetics programme at the Broad Institute, Inc. UKHLS: The UK Household Longitudinal Study is led by the Institute for Social and Economic Research at the University of Essex and funded by the Economic and Social Research Council. The survey was conducted by NatCen and the genome-wide scan data were analysed and deposited by the Wellcome Trust Sanger Institute. Information on how to access the data can be found on the Understanding Society website https://www.understandingsociety.ac.uk/.

## Author contributions

A.-E.F., E.T., M.K., G.D. and E.Z. were involved in the cohort collection. A.-E.F., A.M. and N.W.R. performed phenotype cleaning. L.S., D.S. and J.S. performed bioinformatics analysis. L.S., A.G. and D.S. performed statistical analysis. A.G. and I.T. carried out statistical method development. L.S., A.G., D.S., Y.X. and E.Z. drafted the manuscript.

## Additional information

**Competing interests:** The authors declare no competing financial interests.

