## [Peer Review File · Nature Communications]

Reviewers' comments:

Reviewer #1 (Remarks to the Author):

This is an interesting paper and a demonstration that diversity in the allelic spectrum, such as those in founder populations, can be leveraged for discovery of genotype-phenotype associations. This is not a new idea, however, this is an apparently successful demonstration of the concept.

One thing that is somewhat confusing throughout the paper is the vocabulary around the Cretan population, the MANOLIS and the Pomak cohorts. If these are cohorts, what does it mean to conduct within cohort meta-analysis? Do I understand correctly that genomewide analysis was carried out in N=210 MANOLIS samples and N=734 HELIC Pomack samples? And that corroborative evidence for association was found in these samples? If so, then we need to see the data. It is important to see the nature of the replication evidence the authors claim – in tables 1 and 2, we see only the meta-analyzed results. Particularly for the low-frequency variants in conjunction with such low sample sizes, it would be important to see whether there is statistical corroboration of the findings as the authors claim.

I appreciate the characterization of the enrichment (or not) of variants in this founder population (Figures 1 and 2). For clarity, it would be useful in Figure 2 to either move the key or to put a box around it. The variant densities are inversely correlated with functional importance. A more nuanced discussion of the level of enrichment would be useful, particularly for the coding variants (more likely to be functional). The authors state that they find enrichment of variants with potentially more severe consequences in the lower end of the MAF spectrum, explained by purifying selection. What is evident is a lower representation of alleles in the more common coding alleles, perhaps reflecting purifying selection, but otherwise an enrichment of rarer coding variants likely due to drift. Other variant classes of potential functional significance (regulatory and UTR) are slightly / somewhat enriched similarly across MAF tranches.

There were some other confusing statements. The authors claim in lines 117-119 that excluding the MANOLIS sequences, the signal for HDL is reduced. If the variant is present only in the MANOLIS sequences, then is that not obvious? What is missing here? There is a similar reference in line 138. The authors point to Fig 4c and 4e – its not clear what the reader is supposed to see.

Again, several of the new and associated variants are low frequency in relatively small samples – some sort of corroborative evidence is critical to guard against false positive findings, even at these levels of significance. I would be interested in the minor allele counts for each subset that's analyzed as well as the association stats – betas, SEs. If there was a strategy for internal replication, it needs to be more clearly shown and, preferably, in the body of the paper rather than in supplemental material.

Demonstration of the operational characteristics of a novel approach to meta-analysis (as in Figure 3) seems a distraction in this paper. While it is of interest, it seems this paper is trying to approach too many topics.

Reviewer #2 (Remarks to the Author):

Summary:

Southam and coworkers have reported a characterization of variants from whole-genome sequencing in a Cretan isolated population. Using a WGS-imputation strategy the authors performed GWAS on 19

cardiometabolic traits and identified 9 already know signals and 8 novel associations replicated mostly in the same population. They developed a new method to perform meta-analysis taking into account for non-independence of individuals between datasets. In total the objective of this study is interesting and important and the results are potentially of note. However, below are some concerns for the authors' consideration.

1. The main comment concerns the replication of novel associations. In particular, three signals (DBP, FGBMIadj , and WBC) in Pomak and one signal (WHR) in Manolis were internally replicated even though the top SNPs are not unique to the analyzed population. Indeed, in these associations the best SNP has, in EUR and in the overall 1000G sample, a MAF similar or greater than that observed in the isolates. Have the authors tried to replicate in Manolis associations found in Pomak and vice versa? The authors should justify why these associations were detected only in one isolate. However, a more detailed description of the above associations (including the allele frequency of each associated SNP in reference populations) should be reported in the corresponding paragraph of results.

2. The HGB signal has a very low allele frequency in Pomak (discovery and replication) and the sample size is quite small. However, the regional plot shows some SNPs in strong LD with the best SNP. The presence of a LD proxy SNP in Manolis should be verified and, if this is the case, it should be used for replication. Also for this association, a more detailed description is needed in the results.

3. The MAC now presented in Table 2 for the entire population sample should be reported in supplementary Table 5 for the sample actually used in each association.

4. In the supplementary Table 5 authors should indicate which sample has been used for discovery and replication respectively. In particular, for FGBMIadj association it seems that the discovery has $N=174$ but is unlikely to have a $pvalue= 7.12 \times 10^{-7}$.

5. The interest of using the method implemented in METACARPA in a typical context such as that observed in Pomak and Manolis needs to be discussed by the authors. Further, the authors inferred correlations between the two datasets of each population (Manolis and Pomak) however, as kinship is usually used to measure relatedness between individuals it would be good to report also the kinship between the two datasets of each population.

Reviewer #3 (Remarks to the Author):

The authors perform genome-wide association studies of 31 quantitative traits in two Greek isolated populations. Isolated populations may be particularly interesting to study the impact of rare variants on quantitative traits, as some variants that are rare in the general population might have increased in frequency in those isolated populations, leading to a gain in power for association tests. The authors used a two step strategy for the MANOLIS population, one of their two isolated populations: they first sequenced around 250 individuals of their isolates, which are then used as a reference panel to impute genotypes on the other individuals. The main results of the paper are :

- the characterization of the variation landscape in the MANOLIS population
- the identification of 17 genome-wide significant signals, including 8 novel signals (before Bonferroni correction for the number of traits considered). Two of the novel signals are in loci previously reported
- the development of an approach allowing to perform meta-analysis of studies when there is overlap/relatedness across studies.

1) My main concern regarding the paper is the replication of the new association signals

a) The authors report the likelihood ratio test pvalue. This test might not be suitable here given the low minor count and the low sample size, especially in the replication cohort. According to Supplementary Table 12, there is inflation of the test statistics for some of the phenotypes considered in the MANOLIS replication study. Besides, it seems that some association tests rely on only 1 or 2 carriers in the replication study ? Supplementary Table 5 should report minor allele count in each cohort.

b) Due to the low sample size of the replication cohorts, additional results should be provided to convince that the signals are real.

- for example, for variants that are available in public reference panels (including HRC), do the signals replicate in other populations ? Some of the variants in novel signals are quite frequent (rs13382259 and rs6131100, with a frequency of 4%, with a similar frequency in 1000G EUR, or also rs112037309 with a frequency of 7.5%), and it is not clear why those loci were not identified previously in studies with large sample sizes from general populations. The author should report allele frequency in the general population (perhaps 1000G EUR) for all variants that are listed as new findings

- when the variants are unique to the population considered, the replication could be tested at the gene level (note that it is not clear why the authors state that rs145556679 is unique to their population since it is found once in Tuscans)

2) First part of the paper aims at characterizing the variation landscape in the MANOLIS population, in particular by comparing the variants present in MANOLIS to those available in reference populations. This requires a strict quality control (QC) to make sure that the variants are real (in particular for rare variants), especially considering the low depth of the sequencing. I have some remarks regarding the QC:

a) the author do not mention any filtering on genotype quality (GQ). This QC could lead to the filtering of some bad quality heterozygous genotypes, and hence to the filtering of some rare variants

b) what is the corresponding sensitivity threshold of the VQSLOD filtering ?

c) low complexity or repeat regions might be problematic. It would be interesting to have some statistics on the number of variants in those regions and their frequencies/minor allele counts.

d) which QC was applied on the sequenced samples ? The only filtering that is mentioned is ethnicity

e) are there some multi-allelic variants in the WGS sequence ? How were they treated during the imputation and association analysis ?

3) The authors propose an approach to perform meta-analysis of several studies, when some individuals from different studies may be related. It is not clear to me why the authors have chosen a meta-analysis approach in this particular study: they have the raw imputed data for all individuals, and could perform the analysis by adjusting on the batch so as to take into account the fact that the genotyping/imputation was not performed at once. How does this adjustment approach compares to their meta-analysis approach, especially for rare and low frequency variants, in both their simulation and their real data ? It would be useful in the simulations to have as a reference the results of the global analysis (where overlapping samples are removed, and the analysis is performed in the two

studies combined, with adjustment on the study).

Minor points

5) Line 74: 5.81% does not correspond to the number in Supp Table 3

6) Line 252-253: why is there an exclusion based on imputation quality for the WGS data ?

7) In Supplementary Table 6, it is not clear to me how were computed the concordance and PPV (were they computed only for heterozygote genotypes and homozygous rare genotypes ?)

8) Are the 249 sequenced MANOLIS individuals from the reference panel also among the individuals that were imputed ? In this case, how do the imputed genotypes compare to the sequenced genotypes (in particular heterozygote genotypes for rare and low frequency variants) ? In the association tests, did you use the imputed genotypes or the sequenced ones ?

9) Line 98: the total number of traits indicated is 37, while it seems that 31 traits were investigated (supplementary Table 12)

10) The supplementary Table 9 lists rs816463 as a novel variant for DBP, but this variant does not appear in Table 2

REVIEWERS' COMMENTS:

Reviewer #2 (Remarks to the Author):

The authors have addressed my previous comments.

Reviewer #3 (Remarks to the Author):

Thanks to the authors for replying to my previous comments.

1) My main concern remains replication evidence for the newly identified variants, which seems to be very weak for some of them:

- the total sample size is rather low (one novel association is based on the analysis of 2930 individuals, while the other novel associations are based on less than 1700 individuals)
- some datasets used as internal replication have very low sample size. For example, the two variants highlighted in the abstract were identified in the Manolis cohort only, with nominal significance and same direction of the effects in the two Manolis sub-cohorts, but one of those two sub-cohorts is made up of 210 individuals only. In this sub-cohort, there are less than 3 carriers of those variants, meaning that most of the signal comes from only 1 Manolis sub-cohort.

2) METACARPA approach. The authors have included a new analysis in the simulations used to assess the performance of their METACARPA approach. They have simulated data for two cohorts, with some individuals belonging to both cohorts (duplicated individuals). METACARPA performs meta-analysis of the results in the two cohorts, taking into account that some individuals are duplicated across cohorts. In the other analysis, the authors have performed a global analysis, where data from the two cohorts are merged, one individual from each pair of duplicated individual is removed, and the analysis is adjusted on the cohort variable. In their simulations, the authors show that METACARPA is more powerful than the global analysis when more than 2% of the individuals are duplicated. I do not understand this result, as the available information is the same in the two analyses (since the duplicated individuals provide exactly the same information) ? Where does the additional power of METACARPA come from ?

Minor points:

3)

3.a) In their answer to my previous point 2 regarding the variant QC in the dataset used to characterize the variation landscape in Manolis, the authors mention a QC on the imputation info score, which I still do not understand (see also my previous point 6). My understanding is that this imputation info score comes from the imputation of the Manolis WGS sequence data using the UK10K-1000G reference haplotypes to merge the panels: only variants with info score > 0.7 were included in the merged reference panel. Is this the dataset (after exclusion of the 1000G and UK10K samples) that was used to characterize the variation landscape in Manolis ? My understanding was that this part was based on the WGS sequence data prior to imputation, meaning that the filtering on info score was not performed. Hence, it is not clear how the statistics given in Supplementary Figure 3 relate to the variants that were actually considered to characterize the variation landscape in Manolis.

3.b) In Supplementary Figure 3, the legend mentions "post-phasing" but it should "post-filtered" ?

3.c) It is not clear to me what the authors mean by "Variants intercepted" in Supplementary Figure 3 (is it the proportion of variants called in the sequencing data that are available in the chip data ?)

4) p 13, lines-302-303. It is not clear what this sentence means there (sensitivity threshold for Indels and SNPs). Should this sentence be rather placed after lines 310-311 ?

5) p14, lines 318-319: it would be nice to specify that the concordance is computed using the array data as the reference

6) Table 2: the total sample size of the meta-analysis is not the sum of the sample sizes in each internal replication dataset.

Point-by-point response to reviewers' comments

We thank the reviewers for their constructive comments.

Reviewer #1 (Remarks to the Author):

This is an interesting paper and a demonstration that diversity in the allelic spectrum, such as those in founder populations, can be leveraged for discovery of genotype-phenotype associations. This is not a new idea, however, this is an apparently successful demonstration of the concept.

1. One thing that is somewhat confusing throughout the paper is the vocabulary around the Cretan population, the MANOLIS and the Pomak cohorts. If these are cohorts, what does it mean to conduct within cohort meta-analysis? Do I understand correctly that genomewide analysis was carried out in N=210 MANOLIS samples and N=734 HELIC Pomak samples? And that corroborative evidence for association was found in these samples? The MANOLIS and Pomak are two separate isolated cohorts. The MANOLIS cohort participants are from Southern Greece (Crete) and the Pomak cohort participants are from the Xanthi region in the North of Greece. Within each cohort we have 2 sample sets, those genotyped on the Illumina OmniExpress and HumanExome arrays (jointly called OmniExome array) and those genotyped on the Illumina CoreExome array. To clarify this we have moved Supplementary Fig.1, the study design flow chart, into the main text now as Figure 1. We have additionally annotated the 3 different meta-analyses in the METACARPA box to include the terms within-cohort and across cohorts in Figure 1.

We have also expanded the main text here to clarify this:

“The MANOLIS (n = 1,476) and Pomak (n = 1,737) cohorts were each genotyped in two tranches (Fig. 1), leading to a requirement for within-cohort meta-analysis.”

“We investigated 13 cardiometabolic, 9 anthropometric and 9 haematological traits of medical relevance and report here genome-wide significant signals ($P \leq 5.00 \times 10^{-8}$) that replicate within (nominal significance and the same direction of effect for each array in a cohort) or across the isolates studied (nominal significance and the same direction of effect in MANOLIS and Pomak).”

We have also updated the methods section **Prioritisation and Validation** to use the term across cohort rather than 4-way or overall analysis.

2. If so, then we need to see the data. It is important to see the nature of the replication evidence the authors claim – in tables 1 and 2, we see only the meta-analyzed results. Particularly for the low-frequency variants in conjunction with such low sample sizes, it would be important to see whether there is statistical corroboration of the findings as the authors claim.

We have incorporated Supplementary Table 5, which provides details of the replication evidence, into the main text Table 2.

3. I appreciate the characterization of the enrichment (or not) of variants in this founder population (Figures 1 and 2). For clarity, it would be useful in Figure 2 to either move the key or to put a box around it.

We have boxed the legend in Figure 2.

4. The variant densities are inversely correlated with functional importance. A more nuanced discussion of the level of enrichment would be useful, particularly for the coding variants (more likely to be functional). The authors state that they find enrichment of variants with potentially more severe consequences in the lower end of the MAF spectrum, explained by purifying selection. What is evident is a lower representation of alleles in the more common coding alleles, perhaps reflecting purifying selection, but otherwise an enrichment of rarer coding variants likely due to drift. Other variant classes of potential functional significance (regulatory and UTR) are slightly / somewhat enriched similarly across MAF tranches.

We thank the reviewer for their comment, and have rewritten the relevant paragraph in the main text to make it clearer. Rather than finding an evidence of purifying selection, we observe that rare variants in MANOLIS are significantly more likely to be unique to that cohort if they have consequences of functional relevance, for example if they belong to coding and regulatory regions. This is expected when comparing new variants that haven't had time to fully go through purifying selection with older variants that are likely to be shared. Although fold enrichment figures for functional variants in the common MAF category are indeed negative among positions that are unique to MANOLIS, that depletion is not significant and therefore no conclusion can be drawn for variants with MAF >5%.

5. There were some other confusing statements. The authors claim in lines 117-119 that excluding the MANOLIS sequences, the signal for HDL is reduced. If the variant is

present only in the MANOLIS sequences, then is that not obvious? What is missing here? There is a similar reference in line 138. The authors point to Fig 4c and 4e – its not clear what the reader is supposed to see.

We are referring to the signal rather than the variant and have made the following changes to the text to clarify this:

“When MANOLIS sequences are not included in the reference panel a reduced signal is observed at a different variant (Fig. 5c, 5e).”

“a reduced signal for a different variant is detected when MANOLIS sequences are not included in the reference panel (Fig. 5d and 5f).”

We have also rearranged Figure 5 to make this clearer and have expanded the text in the Figure 5 legend to include this information:

“Figure 5: Association results for chr16:70790626 and rs145556679 and lipid levels. a. Heterozygotes for chr16:70790626 exhibit significantly higher HDL levels than homozygotes. **b.** Heterozygotes for rs145556679 exhibit significantly lower TG and VLDL levels than homozygotes. **c.** Regional association plot for chr16:70790626. **d.** To determine if the signals are detected without MANOLIS sequences in the reference panel we conducted imputation using a combined UK10K + 1000 Genomes reference panel; the regional plot shows that the chr16:70790626 signal is captured with a different lead variant and a decrease in significance. **e.** Regional association plot for rs145556679. **f.** Regional association plot for rs145556679 using a combined UK10K + 1000 Genomes reference panel; the same signal is captured with a different lead variant and a decrease in association strength.”

6. Again, several of the new and associated variants are low frequency in relatively small samples – some sort of corroborative evidence is critical to guard against false positive findings, even at these levels of significance. I would be interested in the minor allele counts for each subset that’s analyzed as well as the association stats – betas, SEs. If there was a strategy for internal replication, it needs to be more clearly shown and, preferably, in the body of the paper rather than in supplemental material.

We have added minor allele counts to Table 2, which now also includes information from Supplementary Table 5. We have moved Supplementary Figure 1, the flow chart study design, into the main text, now Figure 1.

7. Demonstration of the operational characteristics of a novel approach to meta-analysis (as in Figure 3) seems a distraction in this paper. While it is of interest, it seems this paper is trying to approach too many topics.

The development of METACARPA was motivated by the analytical challenges encountered when meta-analysing across potentially related cohorts as part of this study, and the p-values we report use METACARPA to correct for potential relatedness between the intra-cohort datasets. The method is novel and of potential wide interest to the community, for example when meta-analysing non-independent population cohorts.

Reviewer #2 (Remarks to the Author):

Summary:

Southam and coworkers have reported a characterization of variants from whole-genome sequencing in a Cretan isolated population. Using a WGS-imputation strategy the authors performed GWAS on 19 cardiometabolic traits and identified 9 already known signals and 8 novel associations replicated mostly in the same population. They developed a new method to perform meta-analysis taking into account for non-independence of individuals between datasets. In total the objective of this study is interesting and important and the results are potentially of note. However, below are some concerns for the authors' consideration.

1. The main comment concerns the replication of novel associations. In particular, three signals (DBP, FGBMIadj, and WBC) in Pomak and one signal (WHR) in Manolis were internally replicated even though the top SNPs are not unique to the analyzed population. Indeed, in these associations the best SNP has, in EUR and in the overall 1000G sample, a MAF similar or greater than that observed in the isolates. Have the authors tried to replicate in Manolis associations found in Pomak and vice versa? The authors should justify why these associations were detected only in one isolate.

We have looked across the HELIC cohorts to see if signals were replicated and we also looked in published GWAS results. We have incorporated our findings into the results paragraphs in the main text.

(1) DBP

"The allele frequency of rs13382259 is lower in the MANOLIS (MAF 0.024) compared with the 1000 Genomes Project EUR populations (MAF 0.05) and the Pomak population (MAF 0.05). The signal is not associated in MANOLIS ($P = 0.53$) and is not present in the genome-wide summary statistics for the International Consortium for Blood Pressure (ICBP)¹⁸. Proxies for rs13382259 ($r^2 > 0.8$) are present in the International HapMap Project data (<http://hapmap.ncbi.nlm.nih.gov>) and three are present in ICBP summary statistics but none were significantly associated with DBP."

(2) FGBMIadj

"The allele frequency of rs6131100 is higher in the MANOLIS (MAF 0.083) and 1000 Genomes Project EUR populations (MAF 0.053) compared to the Pomak population (MAF 0.039). rs6131100 is not associated with FGBMIadj in MANOLIS ($P = 0.91$), and is not present in genome-wide summary data available from the Meta-Analyses of Glucose and Insulin-related traits Consortium (MAGIC) study (www.magicinvestigators.org)¹⁹⁻²¹. One proxy for rs6131100 was present in the International HapMap Project but this did not show evidence of association in the MAGIC genome-wide summary data for FGBMIadj."

(3) WBC

"rs79748197 has a similar frequency in MANOLIS and is not associated with WBC ($P = 0.19$). It has a higher allele frequency in the 1000 Genomes Project EUR population (MAF 0.014). No proxies are present for rs79748197 in the Pomak population and this trait was not examined in the Haemgen RBC study²³."

(4) WHR

“The signal is not associated with WHR in the Pomak population ($P = 0.39$), and has a higher frequency in the Pomak (MAF 0.038) and 1000 Genomes Project EUR populations (MAF 0.014) compared to MANOLIS (MAF 0.01). rs140087759 has no proxies in MANOLIS ($r^2 > 0.8$) and is not present in the WHR GWAS summary statistics from the Genetic Investigation of ANthropometric Traits (GIANT) study (https://www.broadinstitute.org/collaboration/giant/index.php/GIANT_consortium_data_files)¹⁶.”

Replication of signals observed in isolates, especially when low in frequency, can be challenging. However, we have demonstrated internal replication for all signals. It is not surprising that the signals are not associated across cosmopolitan European populations; this has been observed previously for isolates. For example in three different Italian isolates an association with smoking was observed in two of three isolates¹. In the Greenland Inuit, fatty acid associations, attributable to selection due to high fat diet, were also shown to have a large effect on height and weight in this population and even though the lead variants were at reasonable frequency in Europeans (MAF 0.04 and 0.16) there was no evidence to suggest that these variants also had an effect on weight in Europeans². There are multiple reasons why signals in isolates do not replicate in cosmopolitan populations, for example differences in LD which may be indicative that the lead variant is not causal, environmental homogeneity and winners curse. We have added discussion of this in the Discussion section of the manuscript:

“The remaining five novel associations are present in European populations (1000 Genomes Project EUR MAF ranging from 0.014 to 0.096) but are not significantly associated in GWAS meta-analyses of cosmopolitan populations. This can be due to a number of reasons in addition to winner’s curse, i.e. larger effect sizes in the discovery isolate cohort. For two of these signals, the variant and its proxies are not present in the HapMap reference panel and therefore these variants are not represented in GWAS conducted to date. Three of the associated variants are represented in HapMap and show no evidence of association outside the isolate; this can indicate that the index variant is in LD with the causal variant in the isolate but not in the cosmopolitan population. Furthermore, the effect and therefore the power to detect associations can be increased in isolates due to the environmental and phenotypic homogeneity when compared to other worldwide populations, in addition to extended LD.”

2. However, a more detailed description of the above associations (including the allele frequency of each associated SNP in reference populations) should be reported in the corresponding paragraph of results.

We have added 1000 Genomes Project EUR allele frequencies to the paragraphs and expanded the description of each signal to include the additional lookups we performed.

3. The HGB signal has a very low allele frequency in Pomak (discovery and replication) and the sample size is quite small. However, the regional plot shows some SNPs in strong LD with the best SNP. The presence of a LD proxy SNP in Manolis should be verified and, if this is the case, it should be used for replication.

In the regional plot there are 4 variants that are in strong LD ($r^2 \geq 0.8$) in the Pomak cohort (rs191583256, chr11:5491899, chr11:5633516 and rs748625230). We looked at the

imputation quality and HGB association P value for these variants in the MANOLIS cohort. The lead variant rs557129696 has an extremely low frequency in MANOLIS (MAF 0.0003) and only passed imputation QC in the OmniExome array but it has an imputed MAC of 0.54. rs191583256 failed imputation QC, chr11:5491899 failed imputation QC for MANOLIS CoreExome and has an imputed MAC of 0.998 for OmniExome, rs748625230 is monomorphic in MANOLIS and chr11:5633516 failed imputation QC for MANOLIS CoreExome but passed in MANOLIS OmniExome; there is no evidence of association with this variant in MANOLIS OmniExome.

Cohort	chr:pos	EA	NEA	EAF	effects	beta	se	P
MANOLIS-Pomak	11:5633516	A	G	0.0068	-?--	-1.31328	0.225542	5.79×10^{-09}
MANOLIS	11:5633516	A	G	0.005	-?	-0.250782	0.432733	0.559591
Pomak	11:5633516	A	G	0.00788	--	-1.70959	0.264221	9.78×10^{-11}

4. Also for this association, a more detailed description is needed in the results. This region is intriguing and we have expanded the results section to include the following:

“The G-allele of rs557129696 is not seen in the 1000 Genomes EUR population. Numerous associations with red blood cell traits, anaemia and thalassemias have been linked to this chromosome 11 region²⁵⁻²⁸. We have previously observed an independent signal associated with blood traits in this chromosomal region of extended LD²⁴. Notably, associations between variants in this region and foetal haemoglobin levels²⁹ have been reported in the Sardinian founder population.”

5. The MAC now presented in Table 2 for the entire population sample should be reported in supplementary Table 5 for the sample actually used in each association. We have combined Supplementary Table 5 and Table 2 and added the MAC.

6. In the supplementary Table 5 authors should indicate which sample has been used for discovery and replication respectively. In particular, for FGBMladj association it seems that the discovery has $N=174$ but is unlikely to have a $pvalue= 7.12 \times 10^{-7}$. We have now combined Supplementary Table 5 with Table 2. Our selection criteria were that the meta-analysis had to be genome-wide significant with nominal significance in all contributing strata. We used an independent genotyping technology (Sequenom genotyping) to validate the genotypes and association signal for all reported findings, including the FGBMladj signal for which the Sequenom genotyping GEMMA association for Pomak OmniExome was $P = 3.43 \times 10^{-6}$ with 130 samples and Pomak CoreExome $P = 1.96 \times 10^{-5}$ with 498 samples, please refer to Supplementary Table 5 for the METACARPA Sequenom result.

7. The interest of using the method implemented in METACARPA in a typical context such as that observed in Pomak and Manolis needs to be discussed by the authors. Further, the authors inferred correlations between the two datasets of each population (Manolis and Pomak) however, as kinship is usually used to measure relatedness between individuals it would be good to report also the kinship between the two datasets of each population.

METACARPA uses tetrachoric correlation to infer potential sample overlap or relatedness from summary statistics. This measure expresses overlap between cohorts as the fraction of samples that would overlap if both studies were the same size. When participant genotype data are available, kinship coefficients are a more direct and accurate method of measuring sample relatedness. We now report the average between-dataset kinship in addition to tetrachoric p-value correlations.

When genotype data are available, as they are here, researchers have the option of performing a mega-analysis, removing inferred overlapping samples and adding study provenance as a covariate. We add this design in our simulation scenarios (updated Fig. 4 and Supplementary Fig. 4) and show that it is only more powerful in the presence of little or no overlap. For the level of P value correlations seen between the HELIC datasets, simulation shows that power is similar for both methods, although a mega-analysis better controls the type-I error rate. We further compare the two approaches experimentally in the HELIC datasets (Supplementary Fig. 4 and a new Supplementary Fig. 5 below). Type-I error, as reflected by the median lambda-statistic, is equivalent between a genotype-level meta-analysis, and summary-level approaches, whether they try and correct for sample relatedness (METACARPA) or not (GWAMA).

Supplementary Figure 5: Comparison between three meta-analysis strategies on the HELIC MANOLIS dataset. The trait being meta-analysed is HDL. The top 3 panels are the Manhattan plots for the 3 analysis approaches taken, with the corresponding qq-plots in the lower panels.

Reviewer #3 (Remarks to the Author):

The authors perform genome-wide association studies of 31 quantitative traits in two Greek isolated populations. Isolated populations may be particularly interesting to study the impact of rare variants on quantitative traits, as some variants that are rare in the general

population might have increased in frequency in those isolated populations, leading to a gain in power for association tests. The authors used a two step strategy for the MANOLIS population, one of their two isolated populations: they first sequenced around 250 individuals of their isolates, which are then used as a reference panel to impute genotypes on the other individuals. The main results of the paper are :

- the characterization of the variation landscape in the MANOLIS population
- the identification of 17 genome-wide significant signals, including 8 novel signals (before Bonferroni correction for the number of traits considered). Two of the novel signals are in loci previously reported
- the development of an approach allowing to perform meta-analysis of studies when there is overlap/relatedness across studies.

1. My main concern regarding the paper is the replication of the new association signals

a) The authors report the likelihood ratio test pvalue. This test might not be suitable here given the low minor count and the low sample size, especially in the replication cohort. We based our association test selection on Ma et al.³, who show that the likelihood ratio test exhibits the best balance between type-I error and power under the null, in particular at low allele counts.

b) According to Supplementary Table 12, there is inflation of the test statistics for some of the phenotypes considered in the MANOLIS replication study. There are 3 traits (CRP, FG and HDL) and that have some inflation (\$\lambda > 1.1\$ ) and these are all associated with the MANOLIS CoreExome data set. HDL is one of the novel signals reported in the manuscript (MANOLIS CoreExome HDL \$\lambda = 1.143\$ ) and therefore we have repeated the HDL association analysis taking into account the inflation. For chr16:70790626 and HDL in MANOLIS CoreExome \$P_{GC} = 0.0681\$ (previously \$P = 0.0576\$ ) and for MANOLIS OmniExome \$P_{GC} = 1.5 \times 10^{-9}\$ (previously \$P = 1.8 \times 10^{-10}\$ ). METACARPA \$P_{GC} = 1.55 \times 10^{-11}\$.

c) Besides, it seems that some association tests rely on only 1 or 2 carriers in the replication study ? Supplementary Table 5 should report minor allele count in each cohort. We have combined Supplementary Table 5 with Table 2 thus moving it to the main text in order to make this more prominent and we have added in the MAC for each cohort.

2. Due to the low sample size of the replication cohorts, additional results should be provided to convince that the signals are real.

for example, for variants that are available in public reference panels (including HRC), do the signals replicate in other populations ? Some of the variants in novel signals are quite frequent (rs13382259 and rs6131100, with a frequency of 4%, with a similar frequency in 1000G EUR, or also rs112037309 with a frequency of 7.5%), and it is not clear why those loci were not identified previously in studies with large sample sizes from general populations. Please also refer to our response to Reviewer #2.1. In addition:

“rs112037309 has a higher frequency in the 1000 Genomes Project EUR population (MAF 0.096) compared with the Pomak (MAF 0.073) and MANOLIS (MAF 0.074) populations. We

were unable to look up this variant in large GWAS studies as weight is not one of the traits included as part of the Genetic Investigation of Anthropometric Traits (GIANT)¹⁶ study.”

3. The author should report allele frequency in the general population (perhaps 1000G EUR) for all variants that are listed as new findings

We have added the 1000 Genome Project EUR minor allele frequency to the results paragraphs.

4. when the variants are unique to the population considered, the replication could be tested at the gene level

The chr16:70790626 variant, associated with TG and VLDL, is unique to the MANOLIS population. This variant is located in intron 11 of the *VAC14* gene. To investigate further we took all of the variants in *VAC14* present in at least 1 transcript based on Ensembl and extracted the TG summary data from the Global Lipid Consortium⁴ and Teslovich⁵ studies. Of the 7015 *VAC14* variants 81 in total were present in the summary data and their association *P* values range from 0.076 to 0.98. The MANOLIS HDL signal index variant is observed once in the 1000 Genomes Project Toscani population. The variant is situated in *DSCAML1* and this gene has been previously associated with lipid levels in the Amish. This is described in the main text. The HGB Pomak population association index variant, rs557129696, is observed once in Columbian samples in the 1000 Genomes Project data. This variant resides in a region with excellent candidate genes (haemoglobin-coding genes (*MBE1* and *MBG1*)) and there are many genes in this region that are implicated with blood cell traits⁶⁻⁹.

5. (note that it is not clear why the authors state that rs145556679 is unique to their population since it is found once in Toscani)

rs145556679 is unique to the MANOLIS sequences in the imputation reference panel. This is because the reference haplotypes from the IMPUTEv2 website (ALL.integrated_phase1_SHAPEIT_16-06-14.nosing) only include variants with MAC >1. We have updated the main text to the following:

“This variant is not seen in any other worldwide cohort in the 1000 Genomes Project except for a single heterozygote reported in Toscani in Italia (TSI) samples (n=107, MAF=0.005) (Supplementary Table 7). However, as singletons were filtered out of the reference WGS data prior to phasing, rs145556679 is only represented in the MANOLIS sequences in the reference panel.”

We also altered the discussion to:

“two lipid and the HGB trait signals we identify are driven by variants unique to the MANOLIS cohort or extremely rare in other worldwide populations.”

6. First part of the paper aims at characterizing the variation landscape in the MANOLIS population, in particular by comparing the variants present in MANOLIS to those available in reference populations. This requires a strict quality control (QC) to make sure that the variants are real (in particular for rare variants), especially considering the low depth of the sequencing. I have some remarks regarding the QC:

a) the author do not mention any filtering on genotype quality (GQ). This QC could lead to the filtering of some bad quality heterozygous genotypes, and hence to the filtering of some rare variants

For variant-level QC, we followed the GATK best practices by filtering based on VQSR, and further filtered variants based on Hardy-Weinberg equilibrium, missingness and IMPUTE info score prior to inclusion in the panel. We did not perform genotype-level QC such as GQ (genotype quality) filtering. However, the two rounds of filtering and phasing ensure good genotype quality, as evidenced when comparing 4x calls at various stages of filtering with the hard-called genotypes from the same individuals' chip data. We present below a series of variant capture and genotype/minor allele concordance plots to demonstrate how our QC steps improve genotype quality.

Raw variant unfiltered calls cover most of the GWAS chip positions (Figure 1 below), but the minor allele concordance is below 90% across all MAF categories (blue curve below). In Figures 1,2 and 3 below the right y-axis (Variants Intercepted) refers to the bars and the left y-axis (Concordance) refers to the curves.

Figure 1: Variants called and concordance, 4x calls vs. GWAS (unfiltered)

A first round of variant-level QC is performed using VQSR, which mostly removes rare variants and marginally improves genotype quality (Figure 2 below).

Figure 2: Variants called and concordance, 4x calls vs GWAS (VQSR-filtered)

This is followed by Hardy-Weinberg and missingness exclusion criteria (missingness >3% and HWE $P < 1.00 \times 10^{-4}$). Then after phasing, poorly imputed variants (INFO score <0.7) are filtered out. The variants removed by these two rounds of variant-level QC mostly belong to the rare end of the allele spectrum (half of variants in the 0-0.5% MAF category are filtered out), but some poorly-imputed variants are also removed in the common-frequency categories (purple bars in Figure 3 below). Genotype quality is improved, with an average minor allele concordance of 94.6% for rare (MAF<1%) variants, 96.7% for low-frequency (1%<MAF<5%) variants and 99.6% for common variants (MAF>5%).

Figure 3: Variants called and concordance, 4xcalls vs. GWAS (filtered)

We now include the first and last charts above as a combined Supplementary Fig. 3 for added clarity regarding QC.

b) what is the corresponding sensitivity threshold of the VQSLOD filtering ?

We chose a sensitivity threshold of 90% for INDELS (VQSLOD<3.1159) and a threshold of 94% for SNPs (VQSLOD<5.4079). We have added this to the methods section.

c) low complexity or repeat regions might be problematic. It would be interesting to have some statistics on the number of variants in those regions and their frequencies/minor allele counts.

We added text in the Methods and supplementary material to discuss variant density in low complexity regions (LCR). Briefly, we find that SNP density is much lower in LCR compared to the rest of the genome, and although the allelic makeup of the variants in these regions is different from the average, we do not observe an excess of rare variants in the LCRs.

d) which QC was applied on the sequenced samples ? The only filtering that is mentioned is ethnicity

We applied stringent sample level quality criteria, and found that no samples needed to be excluded based on concordance checks with genotype data, sex checks, mean depth per sample, heterozygous or singleton rate per sample or non-reference allele (NREF) discordance. We have added this to the methods section.

e) are there some multi-allelic variants in the WGS sequence ? How were they treated during the imputation and association analysis ?

Multi-allelic SNPs were excluded prior to inclusion in the panel. We have added this to the methods section.

7. The authors propose an approach to perform meta-analysis of several studies, when some individuals from different studies may be related. It is not clear to me why the authors have chosen a meta-analysis approach in this particular study: they have the raw imputed data for all individuals, and could perform the analysis by adjusting on the batch so as to take into account the fact that the genotyping/imputation was not performed at once. How does this adjustment approach compares to their meta-analysis approach, especially for rare and low frequency variants, in both their simulation and their real data ? It would be useful in the simulations to have as a reference the results of the global analysis (where overlapping samples are removed, and the analysis is performed in the two studies combined, with adjustment on the study).

In order to include only unrelated samples we would need to exclude 41% and 50% of the individuals in MANOLIS and Pomak cohorts, respectively. This would have a major impact on power. To demonstrate this we merged the directly-typed OmniExome and CoreExome genotype data in each cohort, excluded variants with MAF <1% and LD pruned (using $r^2 < 0.2$) and carried out identity by descent analysis in PLINK. The table below shows the number of samples that need to be excluded to create unrelated data for different PI_HAT thresholds.

Cohort	MANOLIS	Pomak
N samples	1476	1737
N variants for IBD	50,272	47,054
PI_HAT	# Sample exclusions	
0.03125	1210	1525
0.0625	1011	1334
0.125	775	1083
0.2	611	868
0.25	556	793
0.5	284	436

We selected a PI_HAT threshold of 0.2 and excluded 611 samples from the MANOLIS cohort and 868 from the Pomak cohort, leaving 865 MANOLIS and 869 Pomak samples for association analysis. We repeated the association tests in these unrelated samples adjusting for array in the imputed data for the 8 reported novel variants using a cohort mega-analysis approach in SNPTEST and for the across-cohort signal (for weight) we meta-analysed the SNPTEST results using GWAMA. The results of the unrelated sample association analysis are shown below alongside the METACARPA results for reference.

Variant	Trait	Cohorts	EA	NEA	METACARPA					Unrelated samples only				
					EAF	beta	se	P	Overall MAC (N)	EAF	beta	se	P	MAC (N)
chr16:70790626	HDL	MANOLIS	T	C	0.006	-1.713	0.254	1.57×10^{-11}	20 (1476)	0.01	-1.9	0.309	2.11×10^{-9}	12 (919)
rs145556679	TG	MANOLIS	C	G	0.013	-1.134	0.17	2.53×10^{-11}	49 (1476)	0.01	-1.1	0.233	3.43×10^{-6}	19 (916)
	VLDL				0.013	-1.131	0.17	2.90×10^{-11}		0.01	-1.1	0.233	3.49×10^{-6}	19 (918)
rs140087759	WHR	MANOLIS	T	C	0.01	1.189	0.209	1.35×10^{-8}	31 (1476)	0.01	1.5	0.094	7.47×10^{-8}	13 (802)
rs13382259	DBP	Pomak	T	A	0.043	0.554	0.1	3.18×10^{-8}	172 (1737)	0.04	0.73	0.145	6.86×10^{-7}	47 (643)
rs6131100	FGBladj	Pomak	A	T	0.037	-0.79	0.139	1.21×10^{-8}	135 (1737)	0.03	-0.5	0.204	9.57×10^{-3}	26 (380)
rs79748197	WBC	Pomak	G	A	0.008	-1.156	0.209	3.00×10^{-8}	31 (1737)	0.01	-1.2	0.312	1.00×10^{-4}	11 (839)
rs557129696	HGB	Pomak	G	T	0.004	-2.027	0.308	4.83×10^{-11}	13 (1737)	0.002	-2.3	0.617	1.79×10^{-4}	3 (832)
rs112037309	Weight	MANOLIS and Pomak	A	G	0.075	0.287	0.0516	2.70×10^{-8}	485 (3213)	0.07	0.31	0.066	2.58×10^{-6}	238 (1646)

As expected, the strength of association is decreased for all novel variants when using only unrelated individuals. This is particularly marked for the HGB signal, with 7 orders of magnitude increase in the *P* value. Only HDL remains genome-wide significant.

We added paragraphs in the main text and Methods to compare the summary-based meta-analysis approach implemented by METACARPA to a global analysis where the within-cohort datasets are added as a covariate. We compared this both using simulation (updated Figure 4 and Supplementary Figure 3, where a systematic comparison to the global analysis is now added), and experimentally on the HELIC datasets. Briefly, when individual level data are available, a global analysis that takes dataset provenance into account and where overlapping samples are removed maintains the type-I error rate at nominal significance. The power of such a global mega-analysis drops dramatically as sample overlap increases, although it is more powerful when no or little overlap is present. When only summary-level statistics are available, METACARPA provides the advantage of a lower false positive rate than a naïve meta-analysis under typical levels of overlap (0-10%), although it fails to control type-I error to nominal levels. Meanwhile, power is conserved compared to the naïve meta-analysis, and is higher than for a sample-level global analysis.

In our experimental data (Supplementary Figure 4), we find little difference in type-I error between the global analysis accounting for dataset provenance, and summary-based meta-analyses, whether they do try and correct for relatedness (METACARPA) or not (GWAMA). Please also refer to the information given in response to Reviewer#2.7.

Minor points

8. Line 74: 5.81% does not correspond to the number in Supp Table 3
Thank you for identifying this, we have corrected the text to 5.49%.

9. Line 252-253: why is there an exclusion based on imputation quality for the WGS data ?

This was done after imputation, prior to inclusion in the panel, and we have corrected this typo in the main text.

10. In Supplementary Table 6, it is not clear to me how were computed the concordance and PPV (were they computed only for heterozygote genotypes and homozygous rare genotypes ?)

Concordance and PPV were calculated as follows:

$$r = \frac{1}{n} \sum_{k=1}^n \frac{\#concordant\ minor\ alleles}{\#minor\ alleles\ in\ reference\ GWAS\ data}$$
$$PPV = \frac{1}{n} \sum_{k=1}^n \frac{\#concordant\ minor\ alleles}{\#minor\ alleles\ in\ sequencing}$$

We have added this information to the main text.

11. Are the 249 sequenced MANOLIS individuals from the reference panel also among the individuals that were imputed ? In this case, how do the imputed genotypes compare to the sequenced genotypes (in particular heterozygote genotypes for rare and low frequency variants) ?

Yes there are 243 MANOLIS samples that overlap between the sequenced MANOLIS and imputed samples. The minor allele concordance is on average 94.2% for rare variants (<1%), 97.4% for low-frequency variants (1%<MAF<5%), and 99.7% for common variants (MAF>5%). We have added this to the supplementary information.

12. In the association tests, did you use the imputed genotypes or the sequenced ones ?
We used the imputed genotypes in the association analysis. We have added this information to the methods section.

13. Line 98: the total number of traits indicated is 37, while it seems that 31 traits were investigated (supplementary Table 12)

Many thanks for spotting this typo, we have changed 19 cardiometabolic traits to 13 cardiometabolic traits.

14. The supplementary Table 9 lists rs816463 as a novel variant for DBP, but this variant does not appear in Table 2

Thank you noticing this, rs816463 has been excluded from Supplementary Table 9 (now Supplementary Table 8).

References

- 1 Sorice, R. *et al.* Association of a variant in the CHRNA5-A3-B4 gene cluster region to heavy smoking in the Italian population. *European journal of human genetics : EJHG* **19**, 593-596, doi:10.1038/ejhg.2010.240 (2011).
- 2 Fumagalli, M. *et al.* Greenlandic Inuit show genetic signatures of diet and climate adaptation. *Science (New York, N.Y.)* **349**, 1343-1347, doi:10.1126/science.aab2319 (2015).
- 3 Ma, C., Blackwell, T., Boehnke, M. & Scott, L. J. Recommended joint and meta-analysis strategies for case-control association testing of single low-count variants. *Genetic epidemiology* **37**, 539-550, doi:10.1002/gepi.21742 (2013).
- 4 Willer, C. J. *et al.* Discovery and refinement of loci associated with lipid levels. *Nature genetics* **45**, 1274-1283, doi:10.1038/ng.2797 (2013).
- 5 Teslovich, T. M. *et al.* Biological, clinical and population relevance of 95 loci for blood lipids. *Nature* **466**, 707-713, doi:10.1038/nature09270 (2010).
- 6 Menzel, S., Garner, C., Rooks, H., Spector, T. D. & Thein, S. L. HbA2 levels in normal adults are influenced by two distinct genetic mechanisms. *British journal of haematology* **160**, 101-105, doi:10.1111/bjh.12084 (2013).
- 7 Milton, J. N. *et al.* Genetic determinants of haemolysis in sickle cell anaemia. *British journal of haematology* **161**, 270-278, doi:10.1111/bjh.12245 (2013).
- 8 Nuinon, M. *et al.* A genome-wide association identified the common genetic variants influence disease severity in beta0-thalassemia/hemoglobin E. *Human genetics* **127**, 303-314, doi:10.1007/s00439-009-0770-2 (2010).
- 9 Uda, M. *et al.* Genome-wide association study shows BCL11A associated with persistent fetal hemoglobin and amelioration of the phenotype of beta-thalassemia. *Proceedings of the National Academy of Sciences of the United States of America* **105**, 1620-1625, doi:10.1073/pnas.0711566105 (2008).

Point-by-point response to reviewers' comments

Reviewer #3 (Remarks to the Author):

3)

3.a) In their answer to my previous point 2 regarding the variant QC in the dataset used to characterize the variation landscape in Manolis, the authors mention a QC on the imputation info score, which I still do not understand (see also my previous point 6). My understanding is that this imputation info score comes from the imputation of the Manolis WGS sequence data using the UK10K-1000G reference haplotypes to merge the panels: only variants with info score > 0.7 were included in the merged reference panel. Is this the dataset (after exclusion of the 1000G and UK10K samples) that was used to characterize the variation landscape in Manolis? My understanding was that this part was based on the WGS sequence data prior to imputation, meaning that the filtering on info score was not performed. Hence, it is not clear how the statistics given in Supplementary Figure 3 relate to the variants that were actually considered to characterize the variation landscape in Manolis.

The following processing steps have been applied to the MANOLIS 4x WGS:

1. VQSR was applied and the dataset was filtered on a per-variant basis according to the VQSLOD thresholds described in the methods.
2. Sample-level QC was applied, 1 ethnic outlier was removed.
3. Further variant-level QC was applied. Multiallelic variants, as well as monomorphics, singletons and indels were removed. Variants were also filtered based on Hardy-Weinberg equilibrium ($P < 10^{-4}$) and missingness ($> 3\%$).
4. Phasing was performed using IMPUTE v.2.
5. Variants with IMPUTE INFO score < 0.7 were removed from the dataset.

The variant set used for describing the variant landscape in MANOLIS (Fig.2, Fig.3 and Supplementary Fig. 1) is the filtered, imputed set (step 5 above). Supplementary Fig. 3 compares this dataset with the unfiltered dataset (prior to step 1 above). We have modified the text in Supplementary Fig. 3 to clearly lay out the datasets that were used for each analysis, which now reads: "Pink-red bars represent raw sequencing variants (pre-filtering), purple bars represent filtered

variants following VQSR, sample and variant-level QC, and phasing (post-filtering).”

3.b) In Supplementary Figure 3, the legend mentions “post-phasing” but it should “post-filtered” ?

We have changed the figure legend of Supplementary Fig. 3 from post-filtered to Imputed, filtered.

3.c) It is not clear to me what the authors mean by “Variants intercepted” in Supplementary Figure 3 (is it the proportion of variants called in the sequencing data that are available in the chip data ?)

Indeed, this is true. The accompanying text for Supplementary Fig.3 has been changed to: “Bars indicate the proportion of variants present in the chip data that were called in the WGS data, per given MAF category.”

4) p 13, lines-302-303. It is not clear what this sentence means there (sensitivity threshold for Indels and SNPs). Should this sentence be rather placed after lines 310-311 ?

Yes, the sentence has been moved to the Variant QC paragraph (line 361): “Post-VQSR, variants were filtered so as to yield a sensitivity threshold of 90% for INDELS ($VQSLOD < 3.1159$) and a threshold of 94% for SNPs ($VQSLOD < 5.4079$).”

5) p14, lines 318-319: it would be nice to specify that the concordance is computed using the array data as the reference

This change has been made to the text. (line 403): “ ...across the MAF spectrum compared to the array data (Supplementary Fig. 5).”

6) Table 2: the total sample size of the meta-analysis is not the sum of the sample sizes in each internal replication dataset.

The overall MAC and N in the last column of table 2 is the total sample size and MAC for all samples in the cohort regardless if they were included in the analysis. To try and make this clearer we have added the word ‘all’ to the legend, which now reads:

“Overall MAC, minor allele count for all samples in the cohorts from which the signal arose, established using the rounded imputed allele

dosages from SNPTEST

(https://mathgen.stats.ox.ac.uk/genetics_software/snptest/snptest.html

);”